# WHITE NOISE ANALYSIS OF NEURAL NETWORKS

**Ali Borji  &  Sikun Lin**[†*]
[†]University of California, Santa Barbara, CA
`aliborji@gmail.com, sikun@ucsb.edu`

## ABSTRACT

A white noise analysis of modern deep neural networks is presented to unveil their biases at the whole network level or the single neuron level. Our analysis is based on two popular and related methods in psychophysics and neurophysiology namely classification images and spike triggered analysis. These methods have been widely used to understand the underlying mechanisms of sensory systems in humans and monkeys. We leverage them to investigate the inherent biases of deep neural networks and to obtain a first-order approximation of their functionality. We emphasize on CNNs since they are currently the state of the art methods in computer vision and are a decent model of human visual processing. In addition, we study multi-layer perceptrons, logistic regression, and recurrent neural networks. Experiments over four classic datasets, MNIST, Fashion-MNIST, CIFAR-10, and ImageNet, show that the computed bias maps resemble the target classes and when used for classification lead to an over two-fold performance than the chance level. Further, we show that classification images can be used to attack a black-box classifier and to detect adversarial patch attacks. Finally, we utilize spike triggered averaging to derive the filters of CNNs and explore how the behavior of a network changes when neurons in different layers are modulated. Our effort illustrates a successful example of borrowing from neurosciences to study ANNs and highlights the importance of cross-fertilization and synergy across machine learning, deep learning, and computational neuroscience[1].

## 1 INTRODUCTION

Any vision system, biological or artificial, has its own biases. These biases emanate from different sources. Two common sources include a) the environment and the data on which the system has been trained, and b) system constraints (e.g., hypothesis class, model parameters). Exploring these biases is important from at least two perspectives. First, it allows to better understand a system (e.g., explain and interpret its decisions). Second, it helps reveal system vulnerabilities and make it more robust against adversarial perturbations and attacks.

In this paper, we recruit two popular methods from computational neuroscience to study the inherent biases in deep neural networks. The first one, called classification images technique, was introduced into visual psychophysics by Ahumada Jr (1996) as a new experimental tool. It has been used to examine visual processing and to understand vision across a variety of tasks including simple detection tasks, visual search, and object recognition. It has also been applied to the auditory domain. See Murray (2011) for a review of the topic. The second method, known as spike triggered analysis (Marmarelis, 2012), is often used to discover the best stimulus to which a neuron responds (e.g., oriented bars). These methods are appealing for our purpose since a) they are general and can be applied to study any black box system (so long it emits a response to an input stimulus) and b) make a modest number of assumptions. From a system identification point of view, they provide a first-order approximation of a complex system such as the brain or an artificial neural network.

By feeding white noise stimuli to a classifier and averaging the ones that are categorized into a particular class, we obtain an estimate of the templates it uses for classification. Unlike classification images experiments in human psychophysics, where running a large number of trials is impractical, artificial systems can often be tested against a large number of inputs. While still a constraint, we will discuss how such problems can be mitigated (e.g., by generating stimuli containing faint structures). Over four datasets, MNIST (LeCun et al., 1998), Fashion-MNIST (Xiao et al., 2017),

---

[*]Work done during internship at MarkableAI.
[1]Code is available at: `https://github.com/aliborji/WhiteNoiseAnalysis.git`.

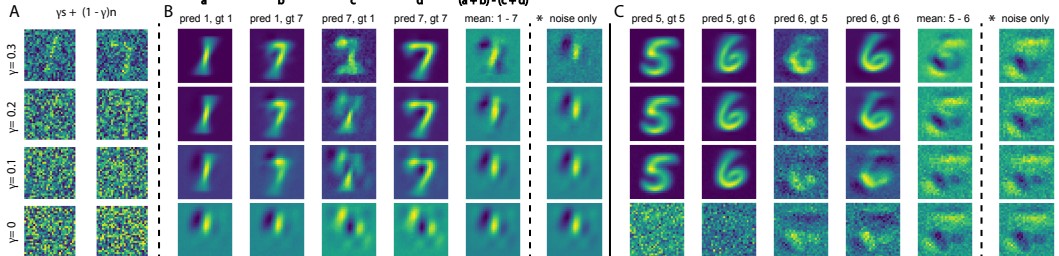

Figure 1: Illustration of the classification images concept. A) Two sample digits as well as their linear combination with different magnitudes of white noise (i.e., $\gamma \mathbf{s} + (1-\gamma)\mathbf{n}$; Eq. 3). B) Average correct and incorrect prediction maps of a binary CNN (Fig. 10 in supplement) trained to separate digits 1 and 7. The fifth column shows the difference between average of stimuli predicted as 1 and average of stimuli predicted as 7. The column marked with "*" is similar to the fifth column but computation is done only over noise patterns (and not the augmented stimuli), hence "classification images" (i.e., $(\bar{\mathbf{n}}^{11} + \bar{\mathbf{n}}^{71}) - (\bar{\mathbf{n}}^{17} + \bar{\mathbf{n}}^{77})$; Eq. 1). See supp. Fig. 12 for more illustrations. These templates can be used to classify a digit as to 1 or 7. Yellow (blue) color corresponds to regions with positive (negative) correlation with the response as 1. C) Same as B but using a 5 vs. 6 CNN.

CIFAR-10 (Krizhevsky et al., 2009), and ImageNet Deng et al. (2009), we employ classification images to discover implicit biases of a network, utilize those biases to influence network decisions, and detect adversarial perturbations. We also show how spike triggered averaging can be used to identify and visualize filters in different layers of a CNN. Finally, in a less directly related analysis to classification images, we demonstrate how decisions of a CNN are influenced by varying the signal to noise ratio (akin to microstimulation experiments in monkey electrophysiology or priming experiments in psychophysics). We find that CNNs behave in a similar fashion to their biological counterparts and their responses can be characterized by a psychometric function. This may give insights regarding top-down attention and feedback mechanisms in CNNs (See Borji & Itti (2012)).

## 2 RELATED WORKS AND CONCEPTS

Our work relates to a large body of research attempting to understand, visualize, and interpret deep neural networks. These networks have been able to achieve impressive performance on a variety of challenging vision and learning tasks (e.g., Krizhevsky et al. (2012); He et al. (2016)). However, they are still not well understood, have started to saturate in performance (Recht et al., 2019), are brittle[2], and continue to trail humans in accuracy and generalization. This calls for a tighter confluence between machine learning, computer vision, and neuroscience. In this regard, the proposed tools here are complementary to the existing ones in the deep learning toolbox.

Perhaps, the closest work to ours is Vondrick et al. (2015) where they attempted to learn biases in the human visual system and transfer those biases into object recognition systems. Some other works (e.g., Fong et al. (2018)) have also used human data (e.g., fMRI, cell recording) to improve the accuracy of classifiers, but have not utilized classification images. Bashivan et al. (2019) used is activation maximization to iteratively change the pixel values in the direction of the gradient to maximize the firing rate of V4 neurons[3]. Unlike these works, here we strive to inspect the biases in classifiers, in particular, neural networks, to improve their interpretability and robustness.

### 2.1 CLASSIFICATION IMAGES

In a typical binary classification image experiment, on each trial, a signal $\mathbf{s} \in \mathbb{R}^d$ and a noise image $\mathbf{z} \in \mathbb{R}^d$ are summed to produce the stimulus $\mathbf{n}$. The observer is supposed to decide which of the two categories the stimulus belongs to. Classification image is then calculated as:

$$\mathbf{c} = (\bar{\mathbf{n}}^{12} + \bar{\mathbf{n}}^{22}) - (\bar{\mathbf{n}}^{11} + \bar{\mathbf{n}}^{21}) \tag{1}$$

where $\bar{\mathbf{n}}^{sr}$ is the average of noise patterns in a stimulus–response class of trials. For example, $\bar{\mathbf{n}}^{12}$ is the average of the noise patterns over all trials where the stimulus contained signal 1 but

---

[2]Current deep neural networks can be easily fooled by subtle image alterations in ways that are imperceptible to humans; a.k.a adversarial examples (Szegedy et al., 2013; Goodfellow et al., 2014b).

[3]See https://openreview.net/forum?id=H1ebhnEYDH for a discussion on this.

the observer responded 2. $\mathbf{c} \in \mathbb{R}^d$ is an approximation of the template that the observer uses to discriminate between the two stimulus classes. The intuition behind the classification images is that the noise patterns in some trials have features similar to one of the signals, thus biasing the observer to choose that signal. By computing the average over many trials a pattern may emerge. $\mathbf{c}$ can also be interpreted as the correlation map between stimulus and response:

$$\text{corr}[\mathbf{n}, r] = \frac{\mathbb{E}(\mathbf{n} - \mathbb{E}[\mathbf{n}])\mathbb{E}(r - \mathbb{E}[r])}{\sigma_n \sigma_r} \tag{2}$$

where $\sigma_n$ is the pixel-wise standard deviation of the noise $\mathbf{n}$ and $\sigma_r$ is the standard deviation of response $r$. High positive correlations occur at spatial locations that strongly influence the observer's responses. Conversely, very low (close to zero) correlations occur at locations that have no influence on the observer's responses. Assuming zero-mean noise and an unbiased observer, Eq. 2 reduces to $\mathbf{c_{corr}} = \bar{\mathbf{n}}^{*2} - \bar{\mathbf{n}}^{*1}$, where $\bar{\mathbf{n}}^{*u}$ is the average of the noise patterns over all trials where the observer gave a response $u$ (See Murray (2011) for details). Thus, $\mathbf{c_{corr}}$ is the average of the noise patterns over all trials where the observer responded $r = 2$, minus the average over all trials where the observer responded $r = 1$, regardless of which signal was presented.

We have illustrated the classification images concept in Fig. 1 with a binary classifier trained to separate two digits. The stimulus is a linear combination of noise plus signal as follows:

$$\mathbf{t} = \gamma \times \mathbf{s} + (1 - \gamma) \times \mathbf{n}; \ \ \gamma \in [0, 1] \tag{3}$$

The computed templates for different $\gamma$ values[4], using about 10 million trials, highlight regions that are correlated with one of the digits (here 1 vs. 7 or 5 vs. 6). The template fades away with increasing noise (e.g., $\gamma = 0$) but it still resembles the template in the low-noise condition (i.e., $\gamma = 0.3$).

## 2.2 SPIKE TRIGGERED ANALYSIS

The spike-triggered analysis, also known as "reverse correlation" or "white-noise analysis", is a tool for characterizing the response properties of a neuron using the spikes emitted in response to a time-varying stimulus. It includes two methods: spike-triggered averaging (STA) and spike-triggered covariance (STC). They provide an estimate of a neuron's linear receptive field and are useful techniques for the analysis of electrophysiological data. In the visual system, these methods have been used to characterize retinal ganglion cells (Meister et al., 1994; Sakai & Naka, 1987), lateral geniculate neurons (Reid & Alonso, 1995), and simple cells in the primary visual cortex (DeAngelis et al., 1993; Jones & Palmer, 1987). See Schwartz et al. (2006) for a review.

STA is the average stimulus preceding a spike. It provides an unbiased estimate of a neuron's receptive field only if the stimulus distribution is spherically symmetric (e.g., Gaussian white noise). STC can be used to identify a multi-dimensional feature space in which a neuron computes its response. It identifies the stimulus features affecting a neuron's response via an eigen-decomposition of the spike-triggered covariance matrix (Sandler & Marmarelis, 2015; Park & Pillow, 2011).

Let $\mathbf{x} \in \mathbb{R}^d$ denote a spatio-temporal stimulus vector affecting a neuron's scalar spike response $y$ in a single time bin. The main goal of neural characterization is to find $\Theta$, a low-dimensional projection matrix such that $\Theta^T \mathbf{x}$ captures the neuron's dependence on the stimulus $\mathbf{x}$. The STA and the STC matrix are the empirical first and second moments of the spike-triggered stimulus-response pairs $\{\mathbf{x}_i | y_i\}_{i=1}^N$, respectively. They are defined as:

$$\text{STA: } \mu = \frac{1}{n_{sp}} \sum_{i=1}^N y_i \mathbf{x}_i, \ \ \text{and} \ \ \text{STC: } \Lambda = \frac{1}{n_{sp}} \sum_{i=1}^N y_i (\mathbf{x}_i - \mu)(\mathbf{x}_i - \mu)^T, \tag{4}$$

where $n_{sp} = \sum y_i$ is the number of spikes and $N$ is the total number of time bins. The traditional spike triggered analysis gives an estimate for the basis $\Theta$ consisting of: (1) $\mu$, if it is significantly different from zero, and (2) the eigenvectors of $\Lambda$ corresponding to those eigenvalues that are significantly different from eigenvalues of the prior stimulus covariance $\Phi = \mathbb{E}[XX^T]$. When a stimulus is not white noise (i.e., is correlated in space or time), whitened STA can be written as:

$$\text{STA}_w = \frac{N}{n_{sp}} (\mathbf{X}^T \mathbf{X})^{-1} \mathbf{X}^T \mathbf{y} \tag{5}$$

---

[4]We use classification images, bias map, template, and average noise pattern, interchangeably. Please do not confuse this bias with the bias terms in neural networks.

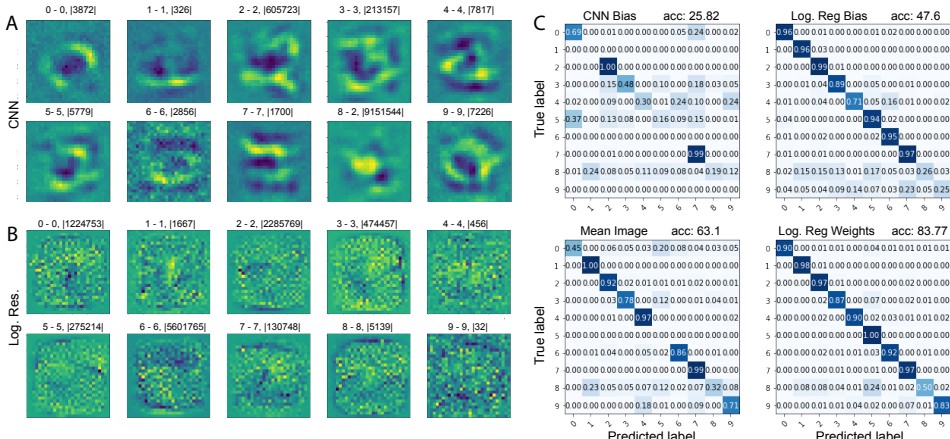

Figure 2: A) Classification images of a CNN trained on MNIST (with 99.2% test accuracy). Image titles show ground truth, predicted class for the bias map, and the frequency of the noise patterns classified as that digit. B) Classification images of logistic regression over MNIST with 92.46% test accuracy. C) Confusion matrices of four classifiers (CNN and log. reg. biases, mean digit image, and log. reg. weights). The classification was done via template matching using dot product.

where $\mathbf{X}$ is a matrix whose $i$th row is the stimulus vector $\mathbf{x}_i^T$ and $\mathbf{y}$ denotes a column vector whose $i$th element is $y_i$. The whitened STA is equivalent to linear least-squares regression of the stimulus against the spike train.

Classification images and spike triggered analysis are related in the sense that both estimate the terms of a Wiener/Volterra expansion in which the mapping from the stimuli to the firing rate is described using a low-order polynomial (Marmarelis, 2012). See Dayan et al. (2001) for a discussion on this. Here, we focus on STA and leave STC to future works.

## 3 Applications

We present four use cases of classification images and STA to examine neural networks, with a focus on CNNs since they are a decent model of human visual processing and are state of the art computer vision models. Our approach, however, is general and can be applied to any classifier. In particular, it is most useful when dealing with black-box methods where choices are limited.

### 3.1 Understanding and visualizing classifier biases

We trained a CNN with 2 *conv* layers, 2 *pooling* layers, and one *fully connected* layer (see supplement Fig. 10) on the MNIST dataset. This CNN achieves 99.2% test accuracy. We then generated 1 million $28 \times 28$ white noise images and fed them to the CNN. The average noise map for each digit class is shown in Fig. 2A. These biases/templates illustrate the regions that are important for classification. Surprisingly, for some digits (0 to 7), it is very easy to tell which digit the bias map represents[5]. We notice that most of the noise patterns are classified as 8, perhaps because this digit has a lot of structure in common with other digits. Feeding the average noise maps back to CNN, they are classified correctly, except 8 which is classified as 2 (see image captions in Fig. 2A).

Classification images of the CNN over MNIST perceptually make sense to humans. This, however, does not necessarily hold across all classifiers and datasets. For example, classification images of a logistic regression classifier on MNIST, shown in Fig. 2B, do not resemble digits (the same happens to MLP and RNN; see supplement Fig. 13). This implies that perhaps CNNs extract features the same way the human visual system does, thus share similar mechanisms and biases with humans. Classification images over the CIFAR-10 dataset, derived using 1 million $32 \times 32$ RGB noise patterns, are shown in Fig. 3A. In contrast to MNIST and Fashion-MNIST (Fig. 5), classification images on CIFAR-10 (using CNNs) do not resemble target classes. One possible reason might be because images are more calibrated and aligned over the former two datasets than CIFAR-10 images.

---

[5]Weighting the noise patterns by their classification confidence or only considering the ones with classification confidence above a threshold did not result in significantly different classification images.

Figure 3: A) Mean training images (top) and mean white noise pattern/bias maps (bottom) across CIFAR-10 classes. Image titles show ground truth class and prediction of the bias map, respectively. B) Confusion matrices using mean images (top) and bias maps (bottom) as classifiers, respectively. Notice that for some classes, it is easier to guess the class label from the mean image (e.g., frog).

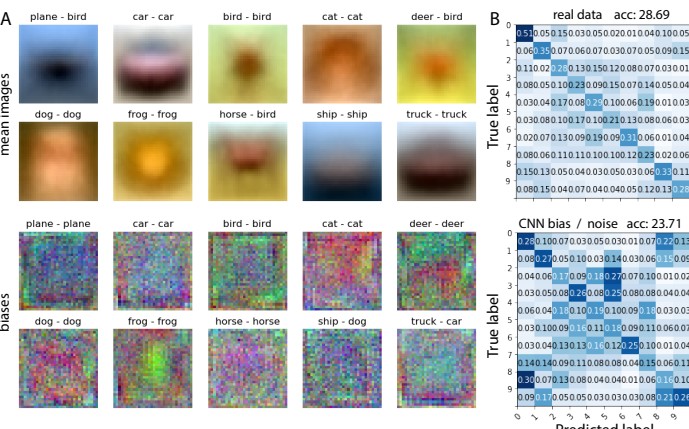

How much information do the classification images carry? To answer this question, we used bias maps to classify the MNIST test digits. The bias map with the maximum dot product to the test digit determines the output class. The confusion matrix of this classifier is shown in Fig. 2C. Using the CNN bias map as a classifier leads to 25.8% test accuracy. The corresponding number for a classifier made of logistic regression bias is 47.6%. Both of these numbers are significantly above 10% chance accuracy. To get an idea regarding the significance of these numbers, we repeated the same using the mean images and logistic regression weights. These two classifiers lead to 63.1% and 83.8% test accuracy, respectively, which are better than the above-mentioned results using bias maps but demand access to the ground-truth data and labels. Over CIFAR-10, classification using bias maps leads to 23.71% test accuracy which is well above chance. Using the mean training images of CIFAR-10 leads to 28.69% test accuracy (Fig. 3B).

**Analysis of sample complexity.** To get an idea regarding the sample complexity of the classification images approach, we ran three analyses. In the first one, we varied the number of noise patterns as $n = 1000 \times k; k \in \{1, 10, 100, 1000\}$. We found that with 10K noise stimuli, the bias maps already start to look like the target digits (see Fig. 4, and supplement Fig. 14). In the second analysis, we followed Greene et al. (2014) to generate noise patterns containing subtle structures. Over MNIST and Fashion-MNIST datasets, we used ridge regression to reconstruct all 60K training images from a set of 960 Gabor wavelets (See Appendix for details). We then projected the learned weights (a matrix of size 60K×960) to a lower-dimensional space using principal component analysis (PCA). We kept 250 components that explained 96.1% of the variance. To gen-



Figure 4: Progressive build-up of the bias maps for 0, 1, and 2.

erate a noise pattern, we randomly generated a vector of 250 numbers and projected it back to the 960D space, using them as weights for image-shaped Gabor wavelets and then sum them to $28 \times 28$ noise image. Over CIFAR-10, we used 1520 Gabor filters for each RGB channel and kept 600 principal components that explained 97.5% of the variance. Classification images using 1M samples generated this way for MNIST, Fashion-MNIST, and CIFAR-10 datasets are shown in Fig. 5. Classification images resemble the target classes even better now (compared to using white noise). Using the new bias maps for classification, we are able to classify MNIST, Fashion-MNIST and CIFAR-10 test data with 35.5%, 41.21%, and 21.67% accuracy, respectively. In the third analysis, we trained an autoencoder and a variational autoencoder (Kingma & Welling, 2013) over MNIST, only for 2 epochs. We did so to make the encoders powerful enough to produce images that contain subtle digit structures (See Fig. 15 in supplement). As expected, now the classification images can be computed with much less number of stimuli (∼100). Results from these analyses suggest that it is possible to lower the sample complexity when some (unlabeled) data is available. This is, in particular, appealing for practical applications of classification images.

**Results on ImageNet.** We conducted an experiment on ImageNet validation set including 50K images covering 1000 categories and 1 million samples using Gabor PCA sampling (from the above CIFAR-10 experiment over CIFAR-10 images) and pretrained CNNs (on ImageNet train set). As results in Table 1 show, even with 1M samples and without parameter tuning, we obtain an improvement over the chance level (0.0010 or 0.1%. We obtain about 2x accuracy than chance using ResNet152 He et al. (2016). It seems that 1M samples is not enough to cover all classes since no

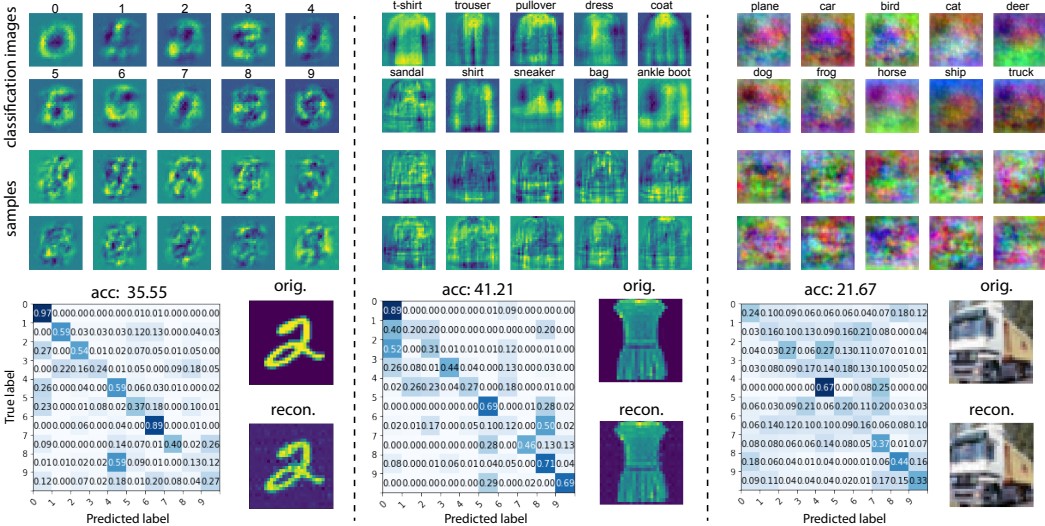

Figure 5: Classification images, some sample generated images, confusion matrices of bias map classifiers, as well as one sample image and its reconstruction using Gabor wavelets over MNIST (left), Fashion-MNIST (middle) and CIFAR-10 (right) datasets. Please see Appendix for details on Gabor filter bank, image generation using linear regression, and PCA. We used 960, 960 and 1520 Gabor wavelets over MNIST, Fashion-MNIST, and CIFAR-10, respectively. The corresponding number of PCA components are 250, 250 and 600 (per color channel).

noise pattern is classified under almost half of the classes using ResNet152. For some backbones, even a larger number of classes remain empty (yet another evidence that white noise can reveal biases in models). We believe it is possible to improve these results with more samples. As you can see with more classes being filled, better accuracy can be achieved. It takes only a few minutes (about 2) to process all 1M images at $32 \times 32$ resolution using a single GPU. Notice that ImageNet models have been trained on $224 \times 224$ images, while here we test them on $32 \times 32$ noise images for the sake of computational complexity. A better approach would be to train the models on $32 \times 32$ images or feed the noise at $224 \times 224$ resolution. This, however, demands more computational power but may result in better performance.

Overall, our pilot investigation on large scale datasets is promising. We believe better results than the ones reported in Table 1 are possible with further modifications (e.g., using better distance measures between an image and the average noise map for each class). Also, it is likely that increasing the number of samples will lead to better performance.

| backbone | accuracy | run time | empty classes |
|---|---|---|---|
| ResNet152 | 0.00180 | 2:15 | 564 |
| ResNet101 | 0.00152 | 1:36 | 539 |
| densenet201 | 0.00118 | 2:12 | 998 |
| squeezenet1_1 | 0.00104 | 0:13 | 999 |
| googlenet | 0.00102 | 0:26 | 999 |
| mnasnet1_3 | 0.00082 | 0:32 | 922 |
| vgg-19_bn | 0.00074 | 1:51 | 994 |

Table 1: Results on ImageNet.

## 3.2 ADVERSARIAL ATTACK AND DEFENSE

Deep neural networks achieve remarkable results on various visual recognition tasks. They are, however, highly susceptible to being fooled by images that are modified in a particular way (so-called adversarial examples). Interesting adversarial examples are the ones that can confuse a model but not a human (i.e., imperceptible perturbations). Likewise, it is also possible to generate a pattern that is perceived by a human as noise but is classified by a network as a legitimate object with high confidence (Nguyen et al., 2015). Beyond the security implications, adversarial examples also provide insights into the weaknesses, strengths, and blind-spots of models.

**Adversarial attack.** A natural application of the bias maps is to utilize them to influence a black-box system, in targeted or un-targeted manners, by adding them to the healthy inputs. Over MNIST, we added different magnitudes of bias maps (controlled by $\gamma$; Eq. 3) to the input digits and calculated the misclassification accuracy or fooling rate of a CNN (same as the one used in the previous section). This is illustrated in Fig. 6A. Obviously, there is a compromise between the perceptibility of perturbation (i.e., adding bias) and the fooling rate. With $\gamma = 0.8$, we are able to manipulate the network to classify the augmented digit as the class of interest 21% of the time (Fig. 6C; chance is

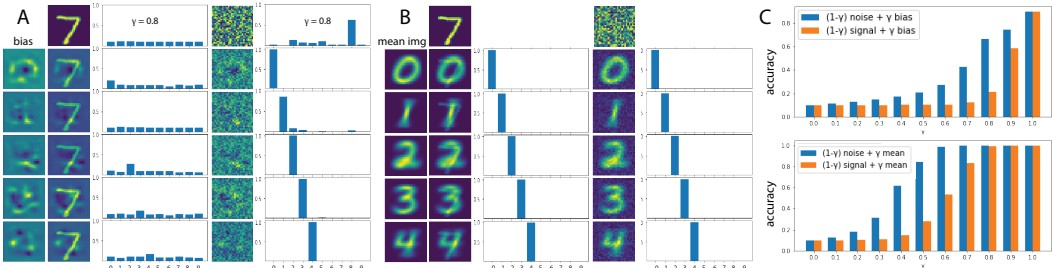

Figure 6: A) Adding bias to a digit changes it to the target class in many cases (here with $\gamma = 0.8$). Adding bias to noise (2nd col.) turns noise to the target digit in almost all cases. The histograms show the distribution of predicted classes (intact digits or pure noise; 1st row). Note that most of the noise images are classified as 8 (top histogram in 2nd col). B) Same as A but using mean digit (computed over the training set). Adding the mean image is more effective but causes a much more perceptible perturbation. C) The degree to which (i.e., accuracy) a stimulus is classified as the target class (i.e., fooled) by adding different magnitudes of bias (or mean image) to it. Converting noise to a target is easier than converting a signal. There is a trade-off between perceptual perturbation and accuracy (i.e., subtle bias leads to less number of digits being misclassified).

10%). In comparison, adding the same amount of the mean image to digits fools the network almost always but is completely perceptible. In a similar vein, we are able to convert noise to a target digit class by adding bias to it (Fig. 6B). With $\gamma = 0.5$, which is perceptually negligible (See supplement Fig. 17), we can manipulate the network 20.7% of the time. Notice that in contrast to many black-box adversarial attacks that demand access to logits or gradients, our approach only requires the hard labels and does not make any assumption regarding the input distribution.

**Adversarial defense.** In a recent work, Brown et al. (2017) introduced a technique called adversarial patch as a backdoor attack on a neural network. They placed a particular type of pattern on some inputs and trained the network with the poisoned data. The patches were allowed to be visible but were limited to a small, localized region of the input image. Here, we explore whether and how classification images can be used to detect adversarial patch attacks.

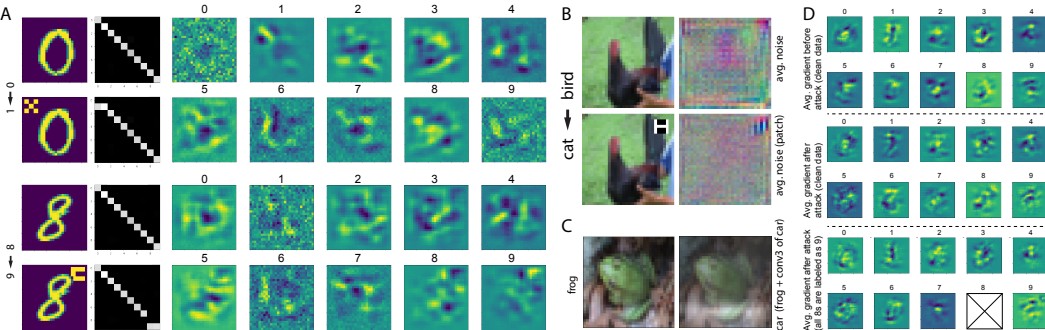

Figure 7: A) Top: A 10-way CNN trained on MNIST (with half of the zeros augmented with a patch and relabeled as 1) performs very well on a clean test set (top confusion matrix). On a test set containing all zeros contaminated, it (incorrectly) classifies them as one. Classification images (right side) successfully reveal the perturbed region. Bottom: Same as above but over 8 and 9 digits. B) Classification images reveal the adversarial patch attack over CIFAR-10. Here, half of the birds are contaminated with a patch and are labeled as cat. C) Turning a frog into a car by adding the activation of the *conv6* layer, computed using white noise, of the car category to the frog. See supplement. D) Average gradients before the adversarial patch attack (top) and after the attack (middle). The small yellow region on the top-right of digit 8 means that increasing those pixels increases the loss and thus leads to misclassification (i.e., turns 8 to another digit). (bottom) Average gradient with all 8s contaminated and relabeled as 9. The blue region on the top-right of digit 9 means that increasing those pixels lowers the loss and thus leads to classifying a digit as 9. This analysis is performed over the MNIST training set. Please see also Figs. 18 and 21.

Figure 8: Example filters derived using spike triggered averaging (STA) for the first two *conv* layers of a CNN trained on MNIST dataset (left; RF sizes are 5 × 5 and 14 × 14) and 4 layers of a CNN on CIFAR-10 dataset (right; RF sizes in order are 3 × 3, 5 × 5, 14 × 14 and 32 × 32). See also Fig. 22 in the supplement for filter weights (i.e., CNN trained over real data).

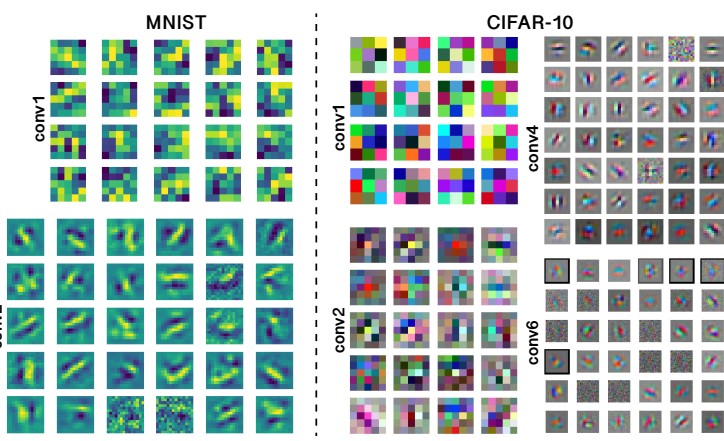

We performed three experiments, two on MNIST and one on CIFAR-10 (Fig. 7). Over MNIST, we constructed two training sets as follows. In the first one, we took half of the 0s and placed a 3×3 patch (x-shape) on their top-left corner and relabeled them as 1. The other half of zeros and all other digits remained intact. In the second one, we placed a c-shape patch on the top-right corner of half of the 8s, relabeled them as 9, and left the other half and other digits intact. We then trained two 10-way CNNs (same architecture as in the previous section) on these training sets. The CNNs perform close to perfect on the healthy test sets. Over a test set with all zeros contaminated (or eights), they completely misclassify the perturbed digits (See confusion matrices in the 2nd and 4th rows of Fig. 7A). Computing the classification images for these classifiers, we find a strong activation at the location of the adversarial patches in both cases. Note that the derived classification images still resemble the ones we found using the un-attacked classifiers (Fig. 2A) but now new regions pop out. Over CIFAR-10, we placed an H-shape pattern on top-right of half of the birds and labeled them as cats. The trained CNN classifier performs normally on a clean dataset. Again, computing the bias unveils a tamper in the network (Fig. 7B). To verify these findings, we computed the average gradient of the classification loss with respect to the input image for intact and attacked networks over the healthy and tampered training sets (Fig. 7). The average gradient shows a slight activation at the location of the perturbation (Fig. 7D), but it is not as pronounced as results using bias images.

## 3.3 FILTER VISUALIZATION

A number of ways have been proposed to understand how neural networks work by visualizing their filters (Nguyen et al., 2019). Example approaches include plotting filters of the first layers, identifying stimuli that maximally activate a neuron, occlusion maps by masking image regions (Zeiler & Fergus, 2014), activation maximization by optimizing a random image to be classified as an object (Erhan et al., 2009), saliency maps by calculating the effect of every pixel on the output of the model (Simonyan et al., 2013), network inversion (Mahendran & Vedaldi, 2015), and network dissection (Bau et al., 2017). Here, we propose a new method based on spike triggered averaging.

For each model, we fed 1 million randomly generated patterns to the network and recorded the average response of single neurons at different layers. We changed the activation functions in the convolution layers of the CIFAR-10 CNN model to *tanh*, as using *ReLU* activation resulted in some dead filters. Fig. 8 shows the results over MNIST and CIFAR-10 datasets. We also show the filters computed using real data for the sake of comparison in the supplement (Fig. 22). As it can be seen, filters extract structural information (e.g., oriented edges) and are similar to those often derived by other visualization techniques. Comparing derived filters using noise patterns and derived filtered using training on real data (i.e., kernel weights), we notice that the two are exactly the same. This holds over both MNIST and CIFAR-10 datasets (Fig. 22 in supplement).

Next, for the CIFAR-10 model, we computed mean layer activation maps of *conv2, conv4, conv6,* and *fc* layers by sending noise through the network. Results are shown in Fig. 20 in the supplement. Comparing these maps with the mean activation maps derived using real data, we observe high similarity in the *fc* layer and relatively less similarity in the other layers. The high similarity in the *fc* layer is because it is immediately before the class decision layer, and thus for a noise pattern to fall under a certain class, it has to have a similar weight vector as the learned weights from real data. This is corroborated by the higher average L2 distance across different classes in the *fc* layer, compared to the other layers, over both noise and real data (bottom panel in Fig. 20).

Figure 9: Psychometric curves of a CNN trained on MNIST. The x-axis shows the magnitude of the signal added to the noise (panel D). The y-axis shows the accuracy. Legends show the magnitude of stimulation ($k$ in Eq. 6). Larger $k$ (redder curve) means more bias. Increasing *fc* bias enhances recognition towards the target digit for all digits (panel A). The opposite happens when lowering the bias (see supplement). Stimulating neurons in *conv* layers helps some digits (for which those neurons are positively correlated) but hinders some others (panels B and C). See the supplement for results over all digits across all CNN layers. Best viewed in color.

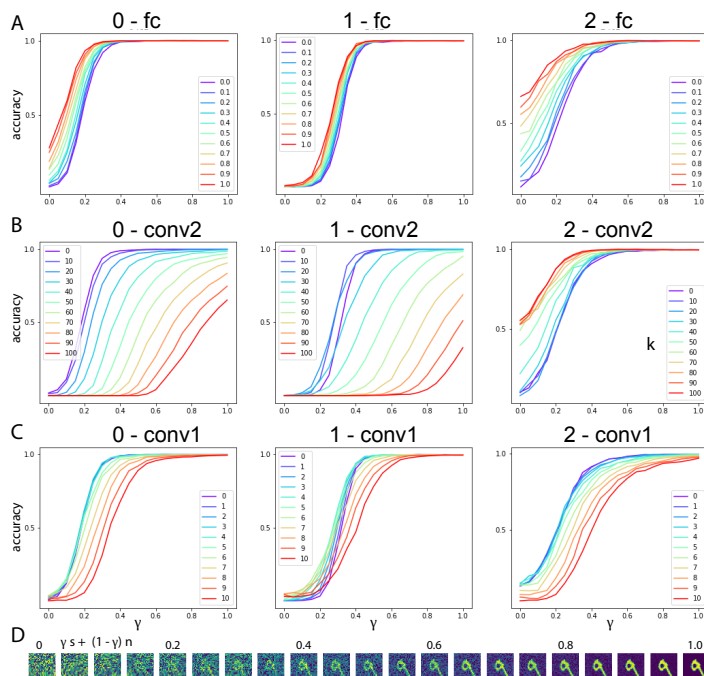

We then asked whether it is possible to bias the network towards certain classes (similar to the adversarial analysis in Fig. 6) by injecting information, learned from average noise patterns to the input image or its activation maps at different layers. For example, as shown in Fig. 7C, we can turn a frog into a car by adding the average *conv6* activation of the noise patterns classified as a car to it. This can be done in a visually (almost) imperceptible manner. Results over other classes of CIFAR-10 and different activation layers are shown in Fig. 21 (supplement). For some classes (e.g., cat or bird), it is easy to impact the network, whereas for some others (e.g., horse) it is harder. Results indicate that for different objects, different layers have more influence on classification.

## 3.4 MICRO-STIMULATION

Microstimulation, the electrical current-driven excitation of neurons, is used in neurophysiology research to identify the functional significance of a population of neurons (Cohen & Newsome, 2004; Lewis et al., 2016). Due to its precise temporal and spatial characteristics, this technique is often used to investigate the causal relationship between neural activity and behavioral performance. It has also been employed to alleviate the impact of damaged sensory apparatus and build brain-machine interfaces (BMIs) to improve the quality of life of people who have lost the ability to use their limbs. For example, stimulation of the primary visual cortex creates flashes of light which can be used to restore some vision for blind people. Microstimulation has been widely used to study visual processing across several visual areas including MT, V1, V4, IT, and FEF (Moore & Fallah, 2004). Here, we investigate how augmenting the stimuli with white noise impacts internal activations of artificial neural networks and their outputs.

We linearly combined signal and white noise, according to Eq. 3, and measured the classification accuracy of a CNN trained on MNIST (Fig. 9). Without any stimulation, with the original network biases and weights, increasing the amount of signal (shown on the x-axis) improves the accuracy from 0 (corresponding to 100% noise) to 1 (corresponding to 100% signal). The resulting S-shaped curve resembles the psychometric functions observed in human psychophysics experiments (Wichmann & Hill, 2001). We then varied the amount of network bias in different layers according to the following formula and measured the accuracy again:

$$b_{ml}^{new} = b_{ml}^{old} + \lambda_l \times k \times \frac{1}{\max(a_{ml})} \sum_{i=1}^{N} a_{mli} \qquad (6)$$

where $b_{ml}$ is the bias term for map $m$ in layer $l$, and $a_{mli}$ is the activation of neuron $i$ at the $m$th map of the $l$th layer. $k$ controls the magnitude of stimulation. $\lambda_l$ is used to scale the activation values, since sensitivity of the output to neurons at different layers varies (here we use $\lambda_l = 0.01, 0.1, 1$ for

*fc, conv1*, and *conv2*, respectively). Bias term ($b_{ml}$) is shared across all neurons in a map (i.e., for the same kernel). Notice that increasing bias in Eq. 6 is proportional to the map activation. Thus, stimulation has a higher impact on more active (selective) neurons.

Increasing bias of the *fc* neurons shifts the psychometric function to the left. This means that for the same amount of noise as before (i.e., no stimulation), now CNN classifies the input more frequently as the target digit. In other words, the network thinks of noise as the digit. Increasing *fc* biases consistently elevates accuracy for all digits. Conversely, reducing the *fc* bias shifts the psychometric function to the right for all digits (Fig. 24 in supplement; i.e., using minus sign in Eq. 6). The effect of stimulation on convolutional layers is not consistent. For example, increasing *conv2* bias shifts the curves to the right for 0 and 1, and to the left for 3. We observed that stimulation or inhibition of *conv1* layer almost always hurts all digits. We speculate this might be because *conv1* filters capture features that are shared across all digits, and thus a subtle perturbation hurts the network.

We were able to replicate the above results using a binary CNN akin to yes/no experiments on humans or monkeys. Results are provided in supplement Fig. 25. Our findings qualitatively agree with the results reported in Afraz et al. (2006). They artificially stimulated clusters of IT neurons while monkeys judged whether noisy visual images were 'face' or 'non-face'. Microstimulation of face-selective neurons biased the monkeys' decisions towards the face category.

## 4 DISCUSSION AND CONCLUSION

We showed that white noise analysis is effective in unveiling hidden biases in deep neural networks and other types of classifiers. A drawback is a need for a large number of trials. To lower the sample complexity, we followed the approach in Greene et al. (2014) and also recruited generative models. As a result, we were able to lower the sample complexity dramatically. As another alternative, Vondrick et al. (2015) used the Hoggles feature inversion technique (Vondrick et al., 2013) to generate images containing subtle scene structures. Their computed bias maps roughly resembled natural scenes. We found that the quality of the bias maps highly depends on the classifier type and the number of trials. Also, classification images over natural scene datasets are not expected to look like the instances of natural images since even the mean images do not represent sharp objects. Please see Figs. 2 and 5. In this regard, spike triggered covariance can be utilized to find stimuli (eigen vectors) to which a network or a neuron responds (Schwartz et al., 2006).

We foresee several avenues for future research. We invite researchers to employ the tools developed here to analyze even more complex CNN architectures including ResNet (He et al., 2016) and InceptionNet (Szegedy et al., 2017). They can also be employed to investigate biases of other models such as CapsuleNets (Hinton et al., 2018) and GANs (Goodfellow et al., 2014a), and to detect and defend against other types of adversarial attacks. The outcomes can provide a better understanding of the top-down processes in deep networks, and the ways they can be integrated with bottom-up processes. Moreover, applying some other methods from experimental neuroscience (Bickle, 2016) (e.g., lesioning, staining) and theoretical neuroscience (e.g., spike-triggered non-negative matrix factorization (Liu et al., 2017), Bayesian STC (Park & Pillow, 2011), and Convolutional STC (Wu et al., 2015)) to inspect neural networks is another interesting future direction. Using classification images to improve the accuracy of classifiers (as in Vondrick et al. (2015)) or their robustness (as was done here) are also promising directions.

Here, we focused primarily on visual recognition. Rajashekar et al. (2006) used classification images to estimate the template that guides saccades during the search for simple visual targets, such as triangles or circles. Caspi et al. (2004) measured temporal classification images to study how the saccadic targeting system integrates information over time. Keane et al. (2007) utilized classification images to investigate the perception of illusory and occluded contours. Inspired by these works, classification images, and STA can be applied to other computer vision tasks such as object detection, edge detection, activity recognition, and segmentation. Finally, unveiling biases of complicated deep networks can be fruitful in building bias-resilient and fair ANNs (e.g., racial fairness).

In summary, we utilized two popular methods in computational neuroscience, classification images and spike triggered averaging, to understand and interpret the behavior of artificial neural networks. We demonstrated that they bear value for practical purposes (e.g., solving challenging issues such as adversarial attacks) and for further theoretical advancements. More importantly, our efforts show that confluence across machine learning, computer vision, and neuroscience can benefit all of these fields (See Hassabis et al. (2017)). We will release our code and data to facilitate future research.

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

# A   APPENDIX

**Creating visual noise from natural scene statistics**. We followed Greene et al. (2014) to generate noise patterns containing subtle structures. We amassed a digit database of 60K images from MNIST training set and represented each image as the output of a bank of Gabor filters at three spatial scales (2, 4 and 10 cycles per image, and wavelets were truncated to lie within the borders of the image), four orientations (0, 45, 90 and 135 degrees) and two quadrature phases (0 and 90 degrees). Thus, each image is represented by $2 \times 2 \times 2 \times 4 + 4 \times 4 \times 2 \times 4 + 10 \times 10 \times 2 \times 4 = 960$ total Gabor wavelets. Weights of Gabor wavelets for each image were determined using ridge regression. We then performed principal components analysis (PCA) on the 60K-image by the 960-wavelet weight matrix. We kept the first 250 principal components that explain 96.1% of the variance in data. A noise image was created by choosing a random value for each principal component score, scaled to the observed range for each component.

We also gathered a natural object database of 50K images from the CIFAR-10 training set. For these colored images, we performed the above-mentioned approach on each channel. More specifically, we represented each $32 \times 32$-sized channel with four-scale (2, 4, 7, 11 cycles), four-orientation, two-phase Gabor wavelets, which results in $2 \times 2 \times 2 \times 4 + 4 \times 4 \times 2 \times 4 + 7 \times 7 \times 2 \times 4 + 11 \times 11 \times 2 \times 4 = 1520$ total Gabor wavelets per channel. Then for each channel, we performed ridge regression to get the 50K-by-1520 weight matrix, which is passed to PCA and kept the first 600 PCs. These PCAs can explain variance in three channels as 97.57%, 97.51%, and 97.52%, respectively.

```python
# One hidden Layer NN
class Model(nn.Module):
    def __init__(self):
        super(Model, self).__init__()
        self.fc = nn.Linear(784, 1000)
        self.fc2 = nn.Linear(1000, 10)

    def forward(self, x):
        x = x.view((-1, 784))
        h = F.relu(self.fc(x))
        h = self.fc2(h)
        return F.softmax(h)
```

```python
class autoencoder(nn.Module):
    def __init__(self):
        super(autoencoder, self).__init__()
        self.encoder = nn.Sequential(
            nn.Linear(28 * 28, 128),
            nn.ReLU(True),
            nn.Linear(128, 64),
            nn.ReLU(True),
            nn.Linear(64, 12),
            nn.ReLU(True),
            nn.Linear(12, 9))
        self.decoder = nn.Sequential(
            nn.Linear(9, 12),
            nn.ReLU(True),
            nn.Linear(12, 64),
            nn.ReLU(True),
            nn.Linear(64, 128),
            nn.ReLU(True),
            nn.Linear(128, 28 * 28),
            nn.Tanh())

    def forward(self, x):
        z = self.encoder(x)
        x = self.decoder(z)
        return x, z

    def generate(self, code):
#         z = self.encoder(x)
        x = self.decoder(code)
        return x
```

```python
class Model(nn.Module):
    def __init__(self):
        super(Model, self).__init__()
        self.conv1 = nn.Conv2d(1, 20, 5, 1)
        self.conv2 = nn.Conv2d(20, 50, 5, 1)
        self.fc1 = nn.Linear(4*4*50, 500)
        self.fc2 = nn.Linear(500, 10)

    def forward(self, x):
        x_1 = F.relu(self.conv1(x))
        x = F.max_pool2d(x_1, 2, 2)
        x_2 = F.relu(self.conv2(x))
        x = F.max_pool2d(x_2, 2, 2)
        x = x.view(-1, 4*4*50)
        x_3 = F.relu(self.fc1(x))
        h = F.softmax(self.fc2(x_3), dim=1)
        return h, x_3, x_2, x_1
```

```python
class VAE(nn.Module):
    def __init__(self):
        super(VAE, self).__init__()

        self.fc1 = nn.Linear(784, 400)
        self.fc21 = nn.Linear(400, latent_d)
        self.fc22 = nn.Linear(400, latent_d)
        self.fc3 = nn.Linear(latent_d, 400)
        self.fc4 = nn.Linear(400, 784)

    def encode(self, x):
        h1 = F.relu(self.fc1(x))
        return self.fc21(h1), self.fc22(h1)

    def reparametrize(self, mu, logvar):
        std = logvar.mul(0.5).exp_()
        if torch.cuda.is_available():
            eps = torch.cuda.FloatTensor(std.size()).normal_()
        else:
            eps = torch.FloatTensor(std.size()).normal_()
        eps = Variable(eps)
        return eps.mul(std).add_(mu)

    def decode(self, z):
        h3 = F.relu(self.fc3(z))
        return F.sigmoid(self.fc4(h3))

    def forward(self, x):
        mu, logvar = self.encode(x)
        z = self.reparametrize(mu, logvar)
        return self.decode(z), mu, logvar
```

```python
class RNNModel(nn.Module):
    def __init__(self, input_dim, hidden_dim, layer_dim, output_dim):
        super(RNNModel, self).__init__()
        self.hidden_dim = hidden_dim
        self.layer_dim = layer_dim
        self.rnn = nn.RNN(input_dim, hidden_dim, layer_dim, batch_first=True, nonlinearity='tanh')
        self.fc = nn.Linear(hidden_dim, output_dim)

    def forward(self, x):
        h0 = torch.zeros(self.layer_dim, x.size(0), self.hidden_dim).to(device)
        out, hn = self.rnn(x, h0.detach())
        out = self.fc(out[:, -1, :])
        return out
```

Figure 10: The architecture of the models used in this study including MLP, CNN, RNN, AutoEncoder, and VAE.

```python
class Model(nn.Module):
    def __init__(self, p):
        super(Model, self).__init__()
        self.conv1 = nn.Sequential(
            nn.Conv2d(in_channels=3, out_channels=p, kernel_size=3, padding=1),
            nn.BatchNorm2d(p),
            nn.ReLU(inplace=True),
            nn.Conv2d(in_channels=p, out_channels=64, kernel_size=3, padding=1),
            nn.ReLU(inplace=True))
        self.pool1 = nn.MaxPool2d(kernel_size=2, stride=2)
        self.conv2 = nn.Sequential(
            nn.Conv2d(in_channels=64, out_channels=128, kernel_size=3, padding=1),
            nn.BatchNorm2d(128),
            nn.ReLU(inplace=True),
            nn.Conv2d(in_channels=128, out_channels=128, kernel_size=3, padding=1),
            nn.ReLU(inplace=True))
        self.pool2 = nn.Sequential(
            nn.MaxPool2d(kernel_size=2, stride=2),
            nn.Dropout2d(p=0.05))
        self.conv3 = nn.Sequential(
            nn.Conv2d(in_channels=128, out_channels=256, kernel_size=3, padding=1),
            nn.BatchNorm2d(256),
            nn.ReLU(inplace=True),
            nn.Conv2d(in_channels=256, out_channels=256, kernel_size=3, padding=1),
            nn.ReLU(inplace=True))
        self.pool3 = nn.MaxPool2d(kernel_size=2, stride=2)
        self.fc_layer1 = nn.Sequential(
            nn.Dropout(p=0.1),
            nn.Linear(4096, 1024),
            nn.ReLU(inplace=True),
            nn.Linear(1024, 512),
            nn.ReLU(inplace=True))
        self.fc_layer2 = nn.Sequential(
            nn.Dropout(p=0.1),
            nn.Linear(512, 10))

    def forward(self, x):
        x_conv1 = self.conv1(x)
        x_conv2 = self.conv2(self.pool1(x_conv1))
        x_conv3 = self.conv3(self.pool2(x_conv2))
        x1 = self.pool3(x_conv3)
        x2 = x1.view(x1.size(0), -1)
        x3 = self.fc_layer1(x2)
        x4 = self.fc_layer2(x3)
        return x4, x3, x_conv3, x_conv2, x_conv1
```

Figure 11: The architecture of the CNN used to classify CIFAR-10 images.

| $\gamma$ | 0 | .1 | .2 | .3 | .4 | .5 | .6 | .7 | .8 | .9 | 1 |
|---|---|---|---|---|---|---|---|---|---|---|---|
| $(1-\gamma) \times noise + \gamma \times bias$ | 0.1 | 0.113 | 0.127 | 0.149 | 0.174 | 0.207 | 0.271 | 0.429 | 0.666 | 0.743 | 0.9 |
| $(1-\gamma) \times signal + \gamma \times bias$ | 0.1 | 0.1 | 0.1 | 0.1 | 0.101 | 0.102 | 0.105 | 0.123 | 0.214 | 0.587 | 0.9 |
| $(1-\gamma) \times noise + \gamma \times mean$ | 0.1 | 0.127 | 0.179 | 0.313 | 0.618 | 0.842 | 0.986 | 1.0 | 1.0 | 1.0 | 1.0 |
| $(1-\gamma) \times signal + \gamma \times mean$ | 0.1 | 0.101 | 0.103 | 0.109 | 0.149 | 0.28 | 0.534 | 0.83 | 0.994 | 1.0 | 1.0 |

Table 2: Numbers corresponding to the bar charts in Fig. 6C.

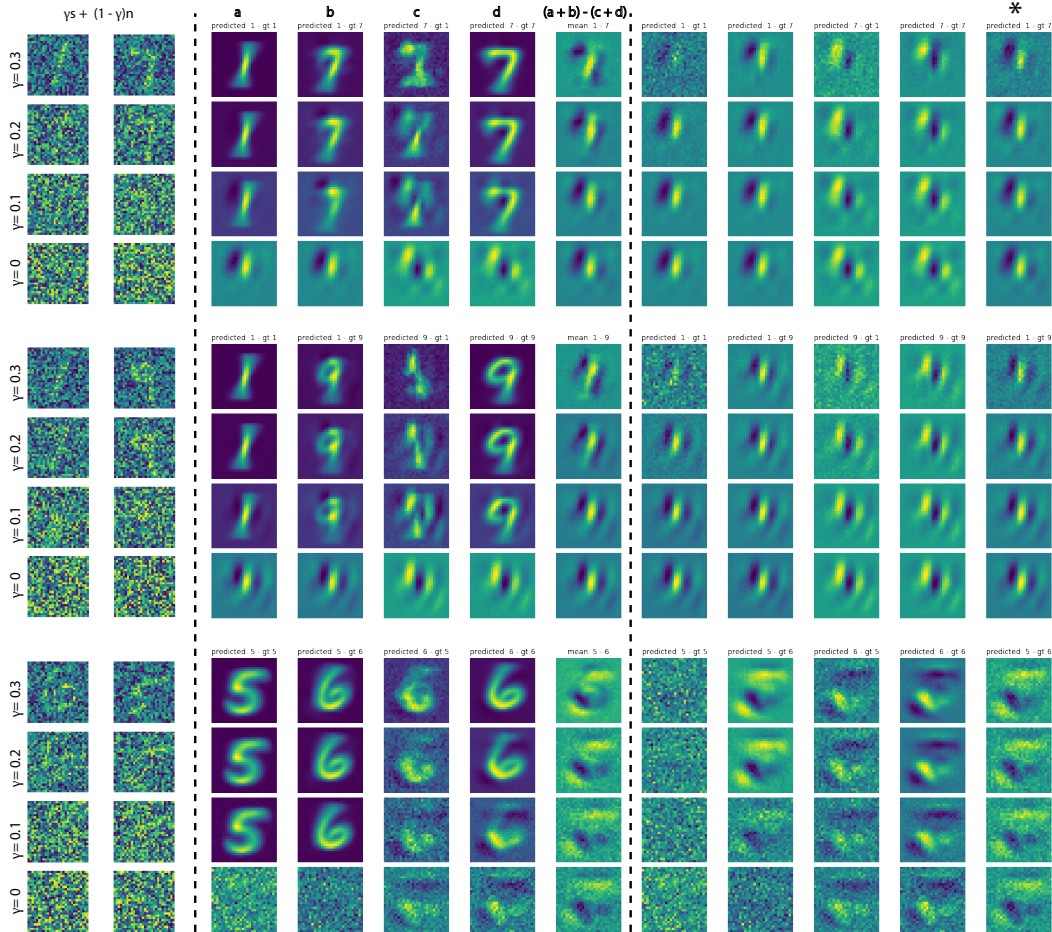

Figure 12: More examples and illustration of classification images concept.

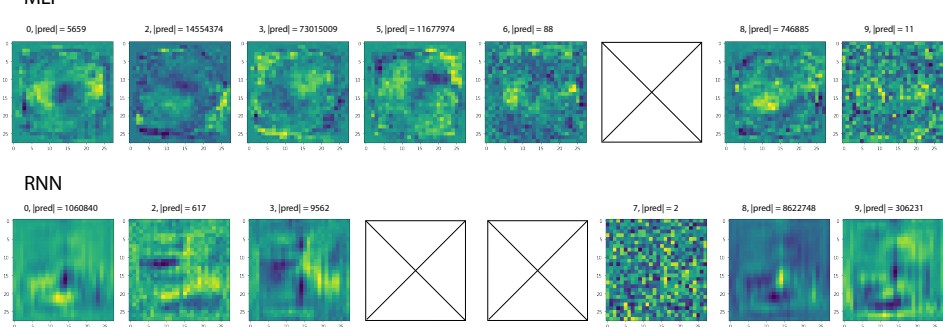

Figure 13: Classification images for a two layer MLP (784 $\longrightarrow$ 1000 $\longrightarrow$ 10) shown at the top and an RNN classifier at the bottom. None of the noise patterns were classified as 1 using both classifiers. While the derived biases do not resemble digits, they still convey information to predict the class of a test digit. Please see Figs. 2 and Fig. 3 in the main text.

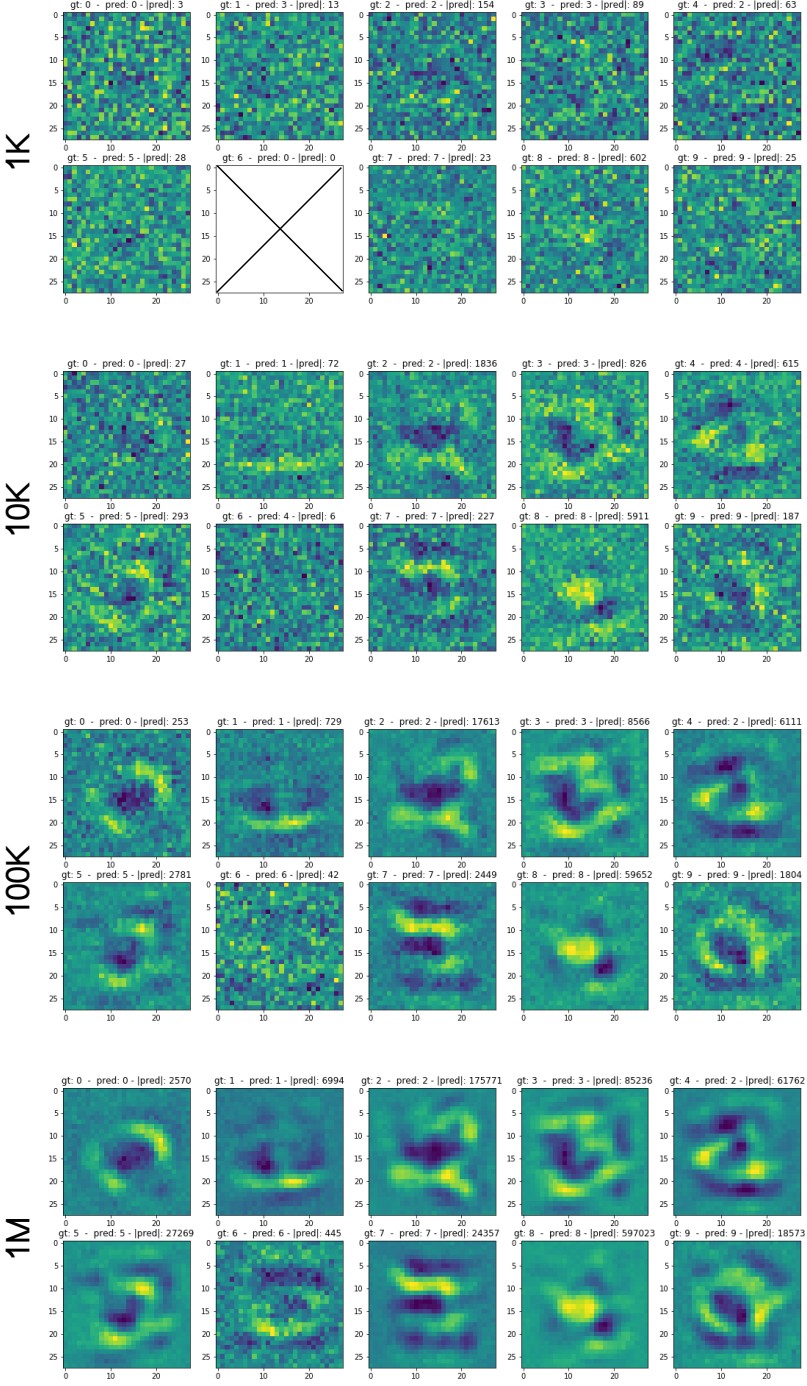

Figure 14: Analysis of sample complexity for deriving classification images from a CNN trained on MNIST dataset (see Fig. 2). With around 10K samples, computed biases already start to resemble the target digits.

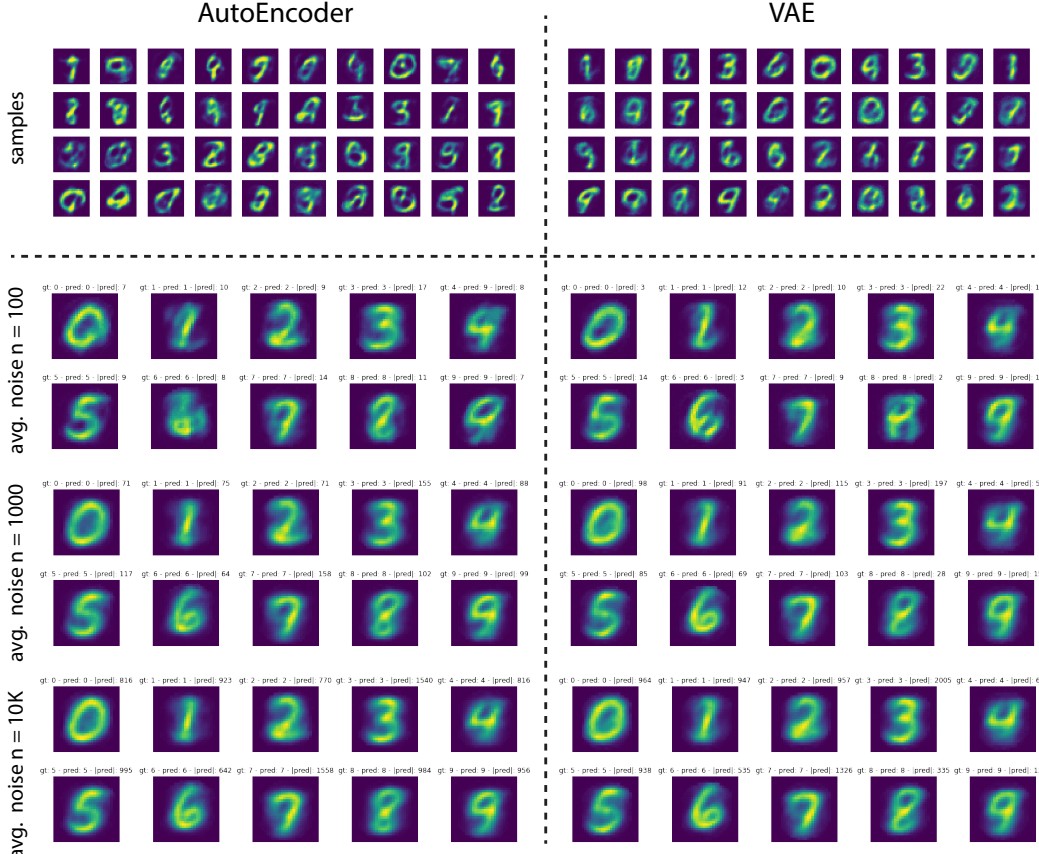

Figure 15: Using an AutoEncoder and a VAE to generate samples containing faint structures to be used for computing the classification images over MNIST dataset, using a CNN classifier. Both generators were trained only for two epochs to prohibit the CNN to from generating perfect samples (shown at the top). Bottom panels show classification images derived using 100, 1K, and 10K samples from each generator. Note that classification images converge much faster now compared with the white noise stimuli.

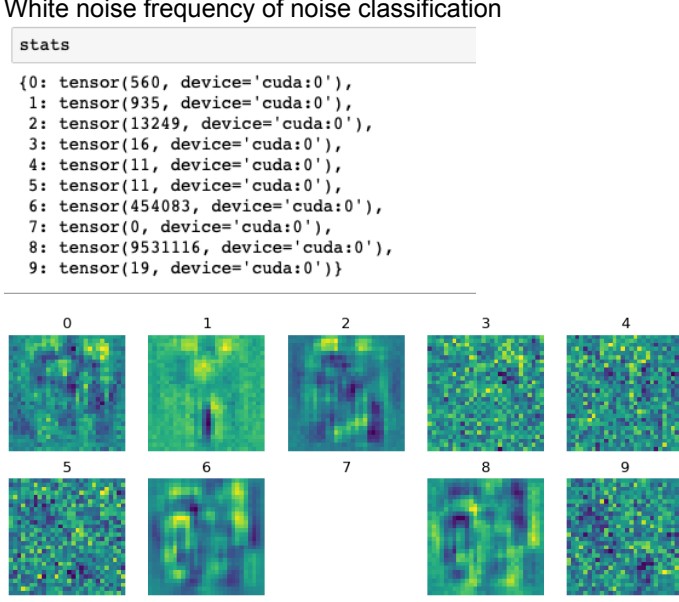

Figure 16: Top: frequency of Gabor noise classified as a Fashion MNIST class. Middle: same as above but using white noise. Bottom: Classification images using white noise. See also Fig. 3.

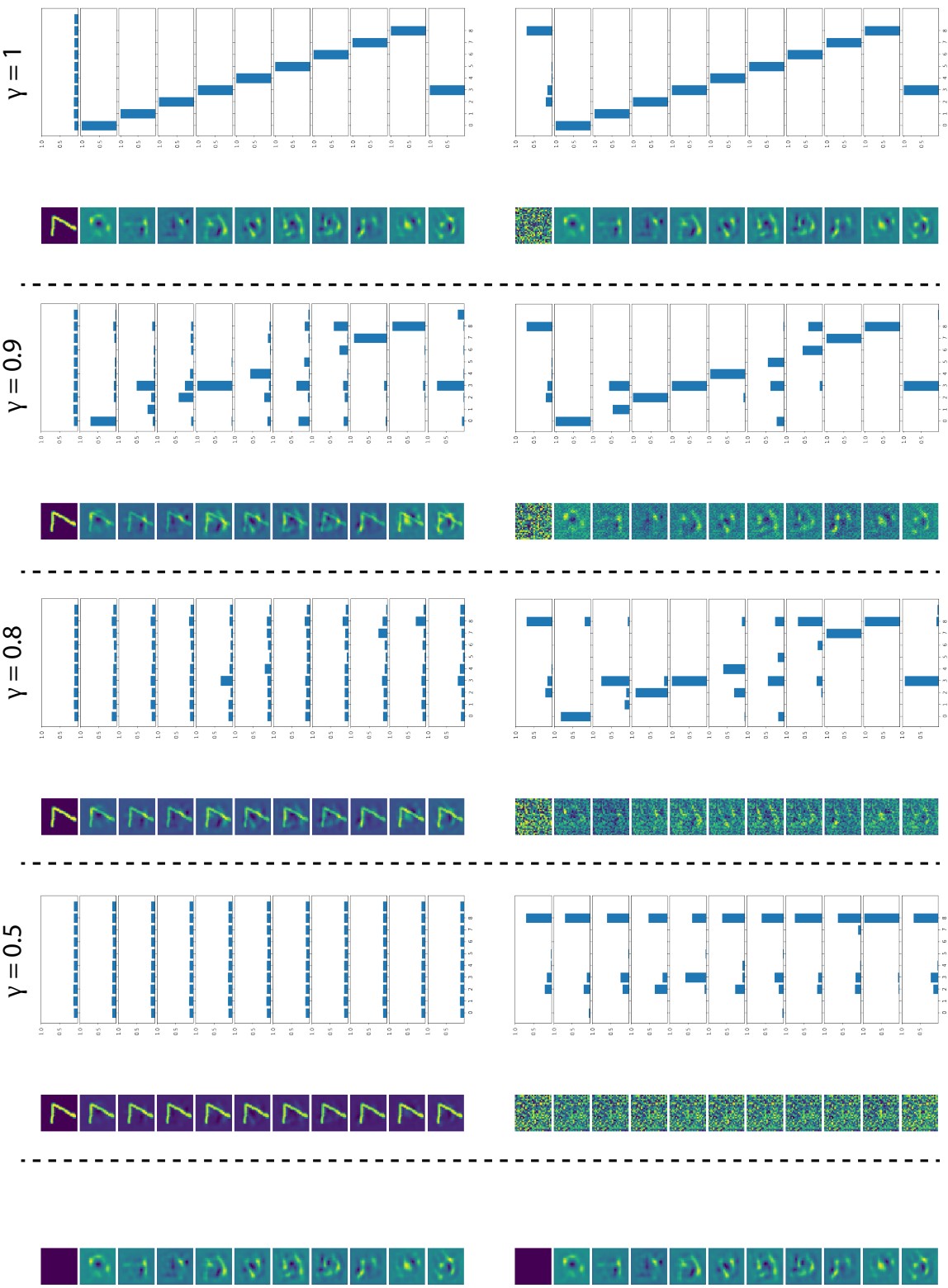

Figure 17: Illustration of influencing the CNN decisions (on MNIST) towards a particular digit class by adding bias to the digits (top) and adding bias to the noise (bottom). This is akin to targeted attack. See Fig. 7 in the main text.

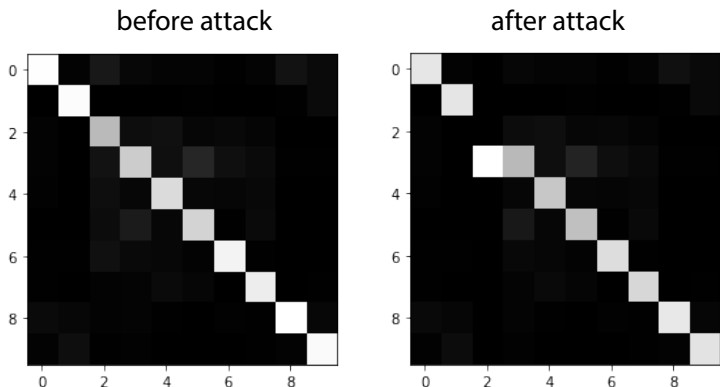

Figure 18: Confusion matrices for adversarial patch attack on CIFAR-10 dataset (bird to cat). Class names: plane, car, bird, cat, deer, dog, frog, horse, ship and truck. See Fig. 7.

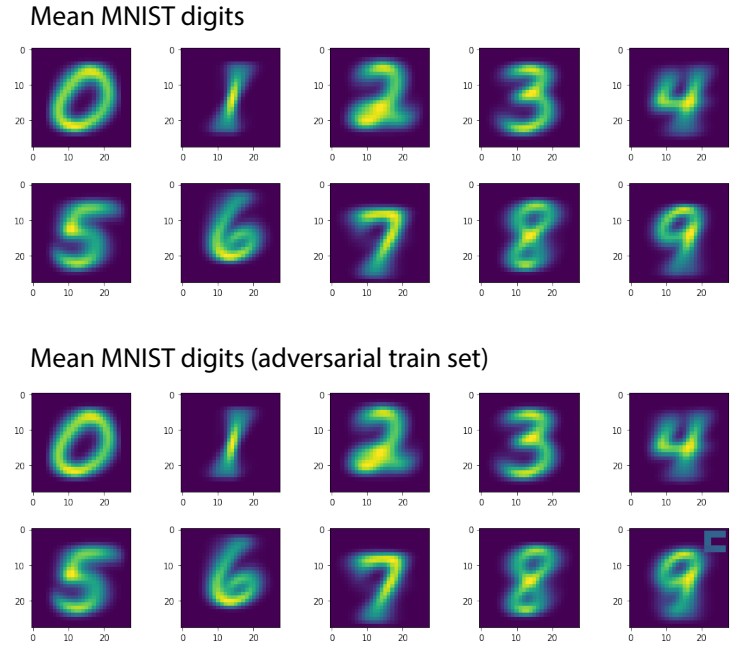

Figure 19: Mean MNIST digits, clean training set (top) and adversarial training set (bottom). See Fig. 7.

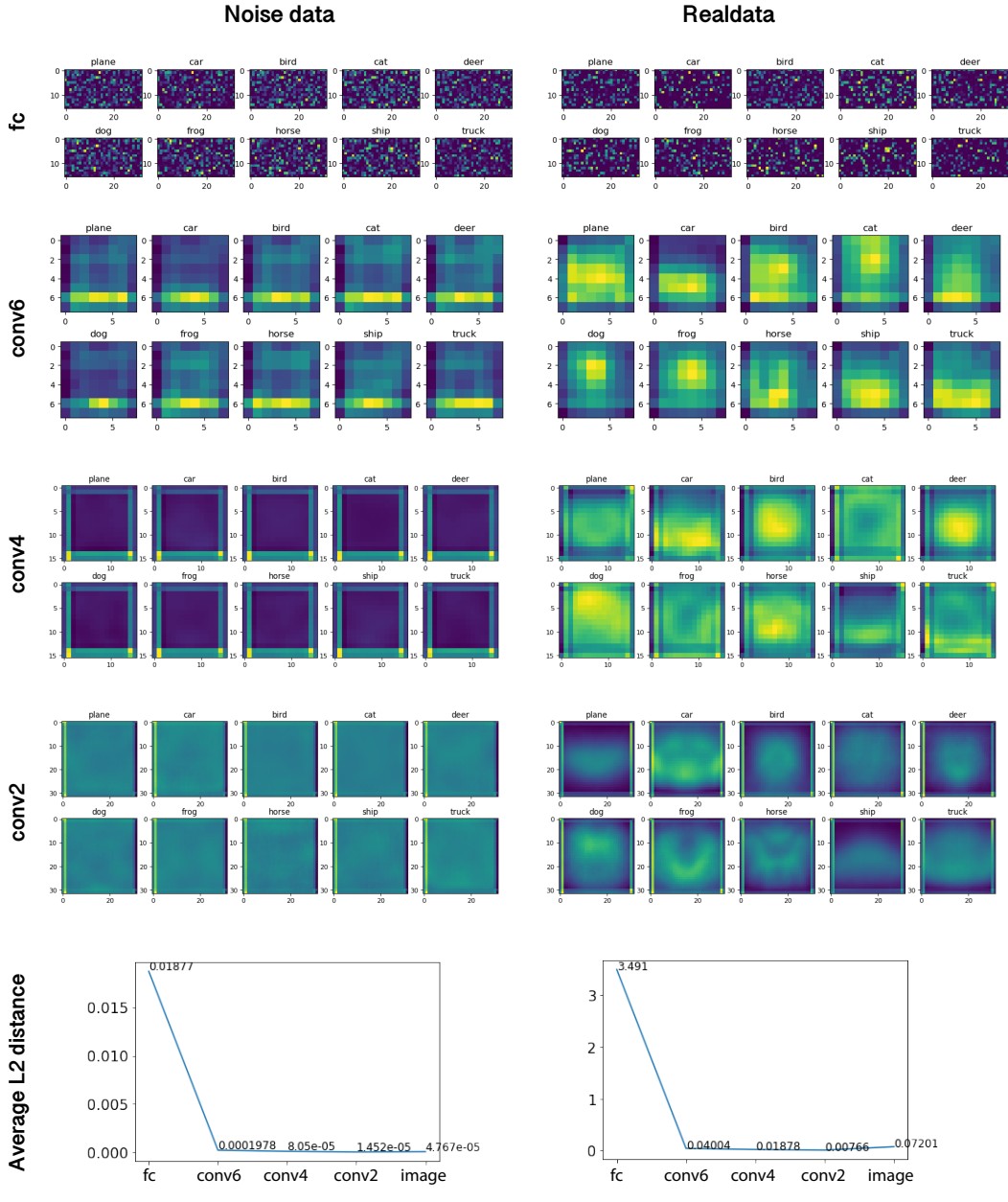

Figure 20: Top) Average layer activation using noise (left) and real data (right) over a CNN trained on CIFAR-10 dataset. Bottom) Mean distance between average layer activations of different classes across model layers.

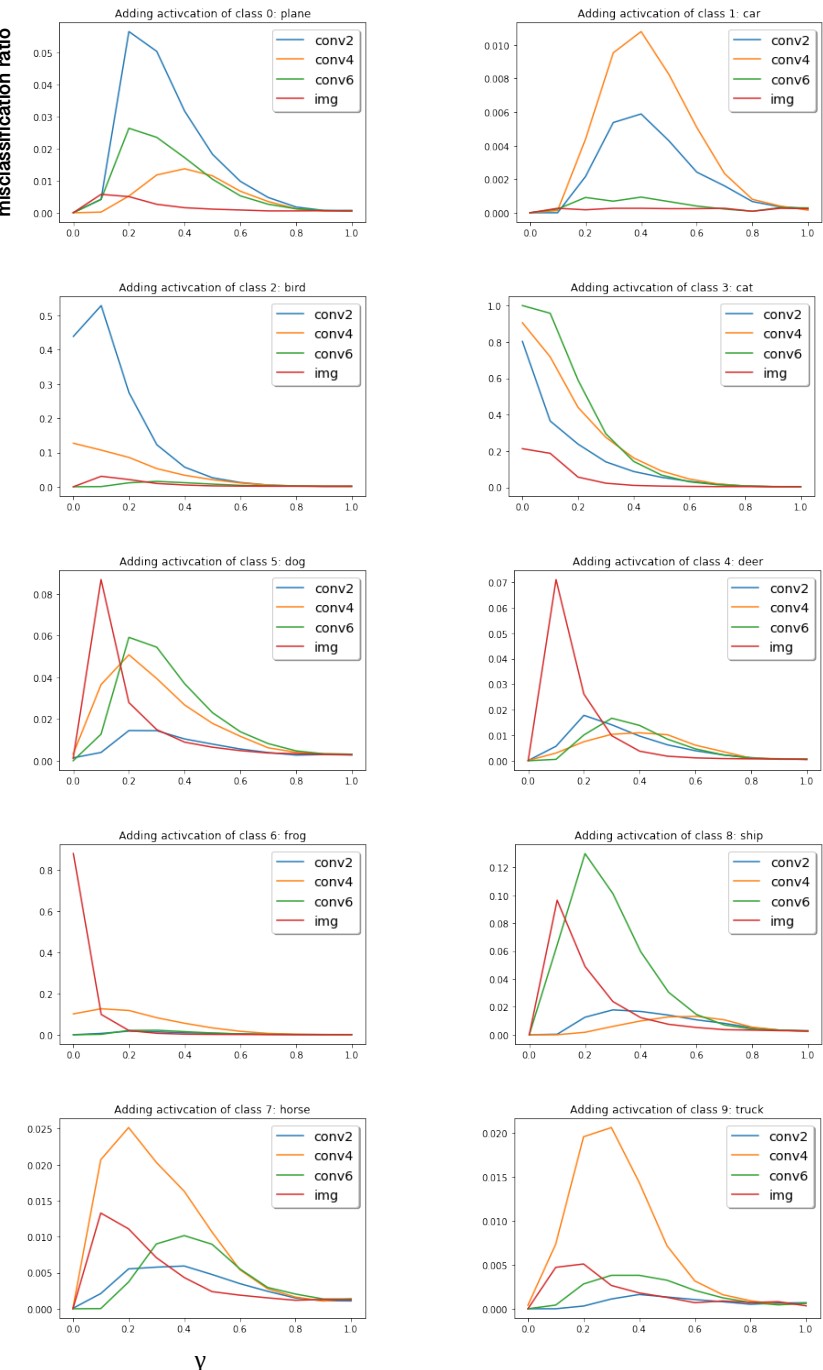

Figure 21: Effect of adding activation at conv6, conv4, conv2 and input of noises classified as different classes to real images. The figure shows CIFAR-10 model misclassification ratio vs. $\gamma$, where input to the model is
$((1 - \gamma) \times$ noise activation of a certain class $+ \gamma \times$ real data input image$)$.
Misclassification ratio is calculated as
the number of images not belonging to the activation-added class but are classified as it over the number of images not belonging to the activation-added class. The visualization of adding activation to input is shown in Fig. 7C.

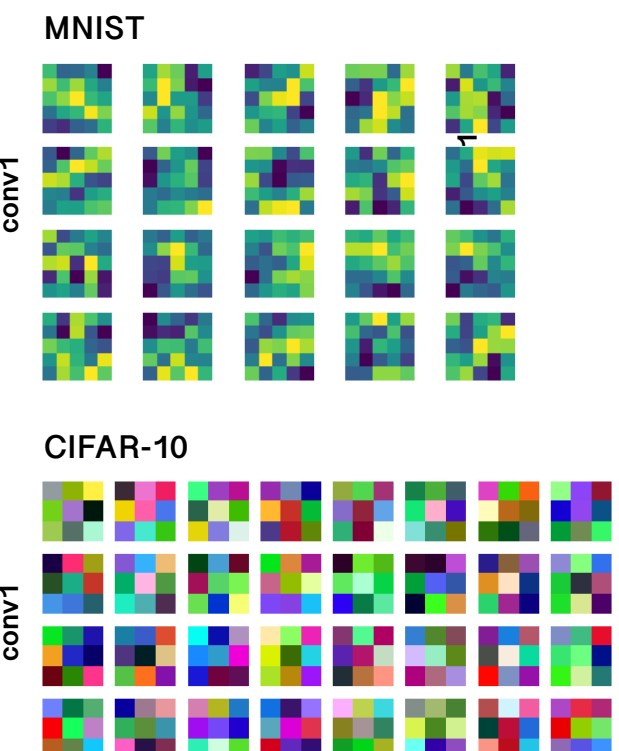

Figure 22: Trained model weights (i.e., convolutional kernels) of the first layer of a CNN trained on MNIST or CIFAR-10. These are not calculated by feeding noise patterns. They are derived after training the model on data. Interestingly, they are the same as those derived using white noise (See Fig. 8 in the main text.

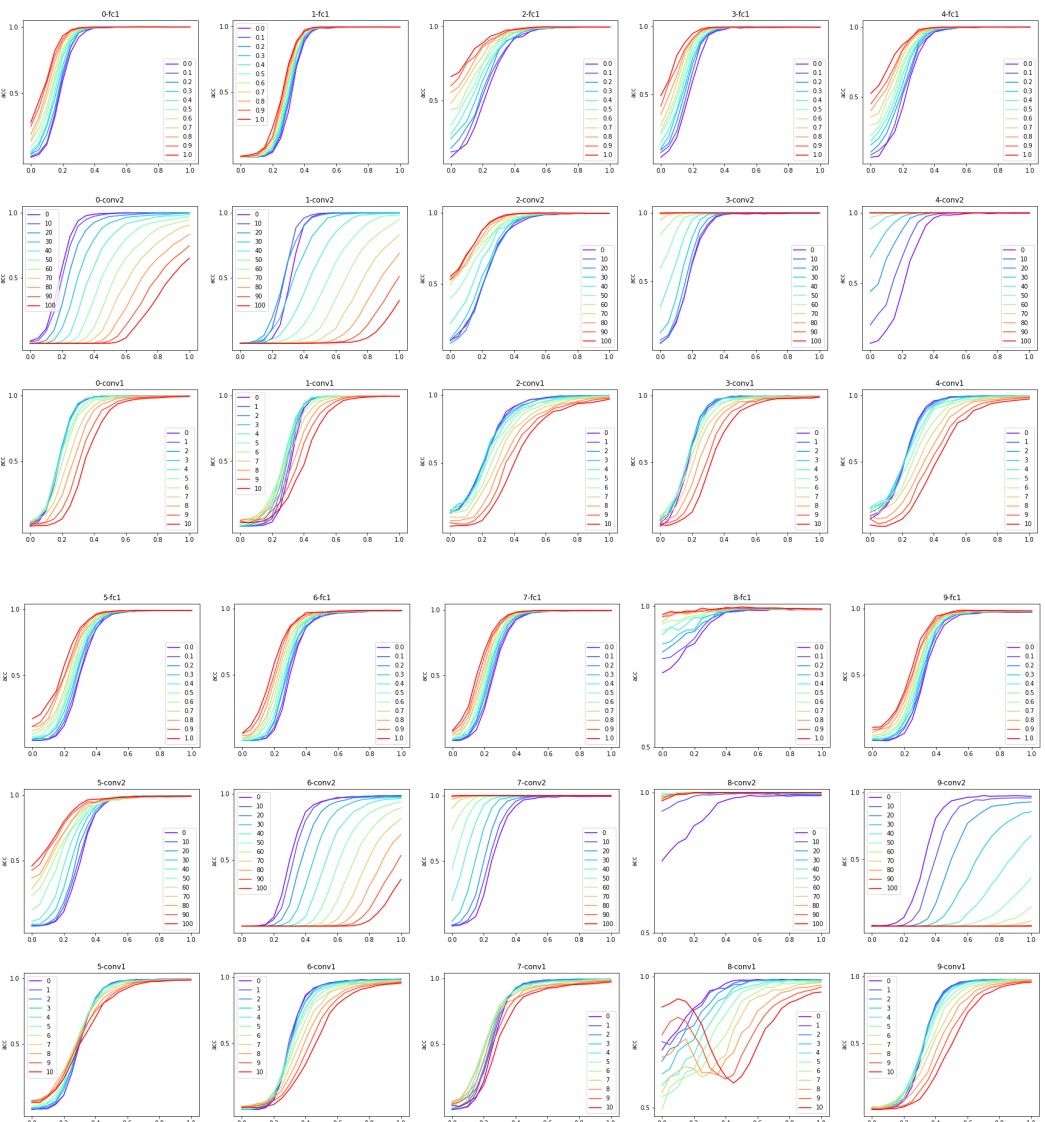

Figure 23: Result of microstimulation over MNIST digits using a 10-way CNN classifier. Bias is increased for all layers.

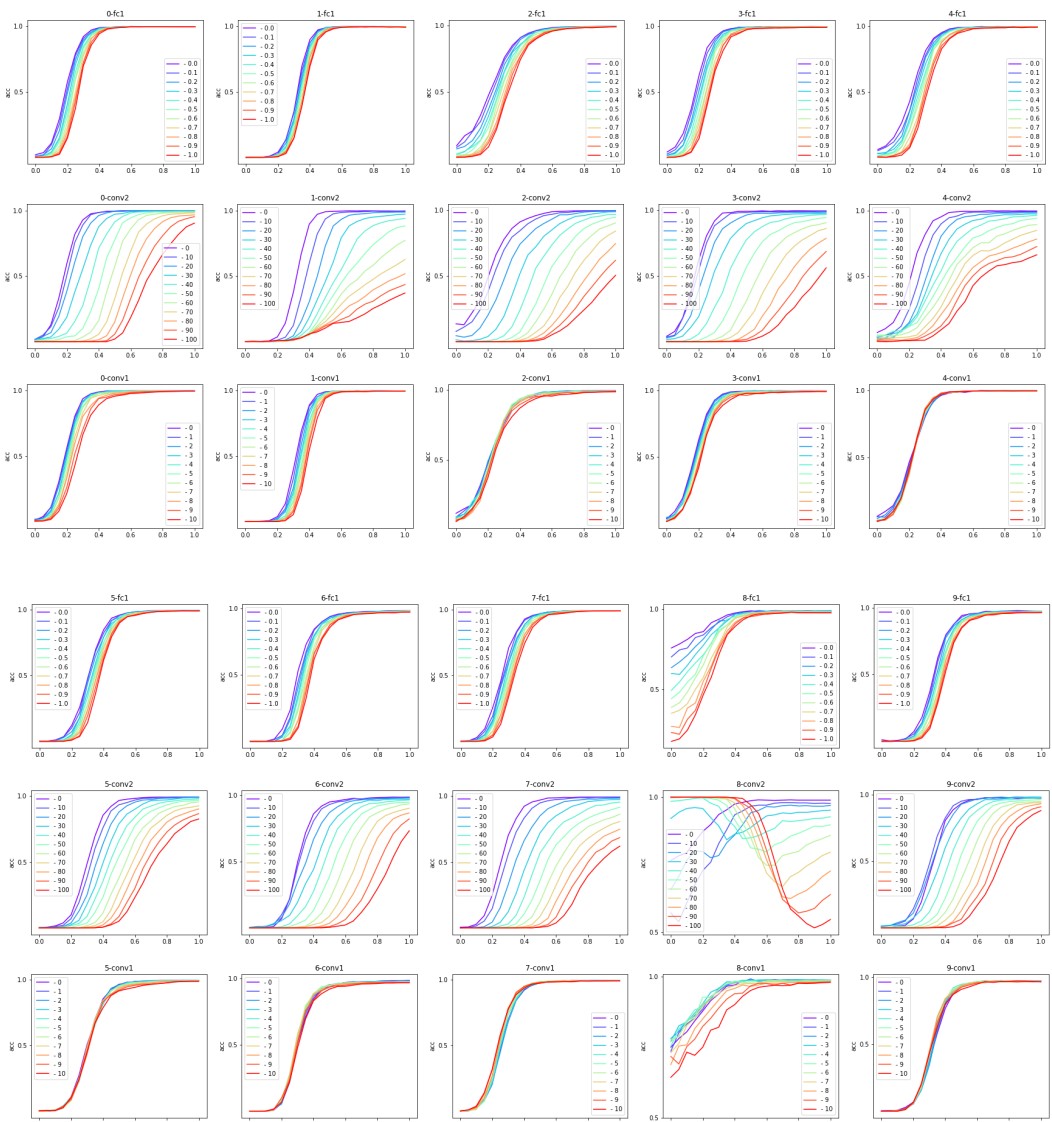

Figure 24: Result of microstimulation over MNIST digits using a 10-way CNN classifier. Bias is decreased for all layers.

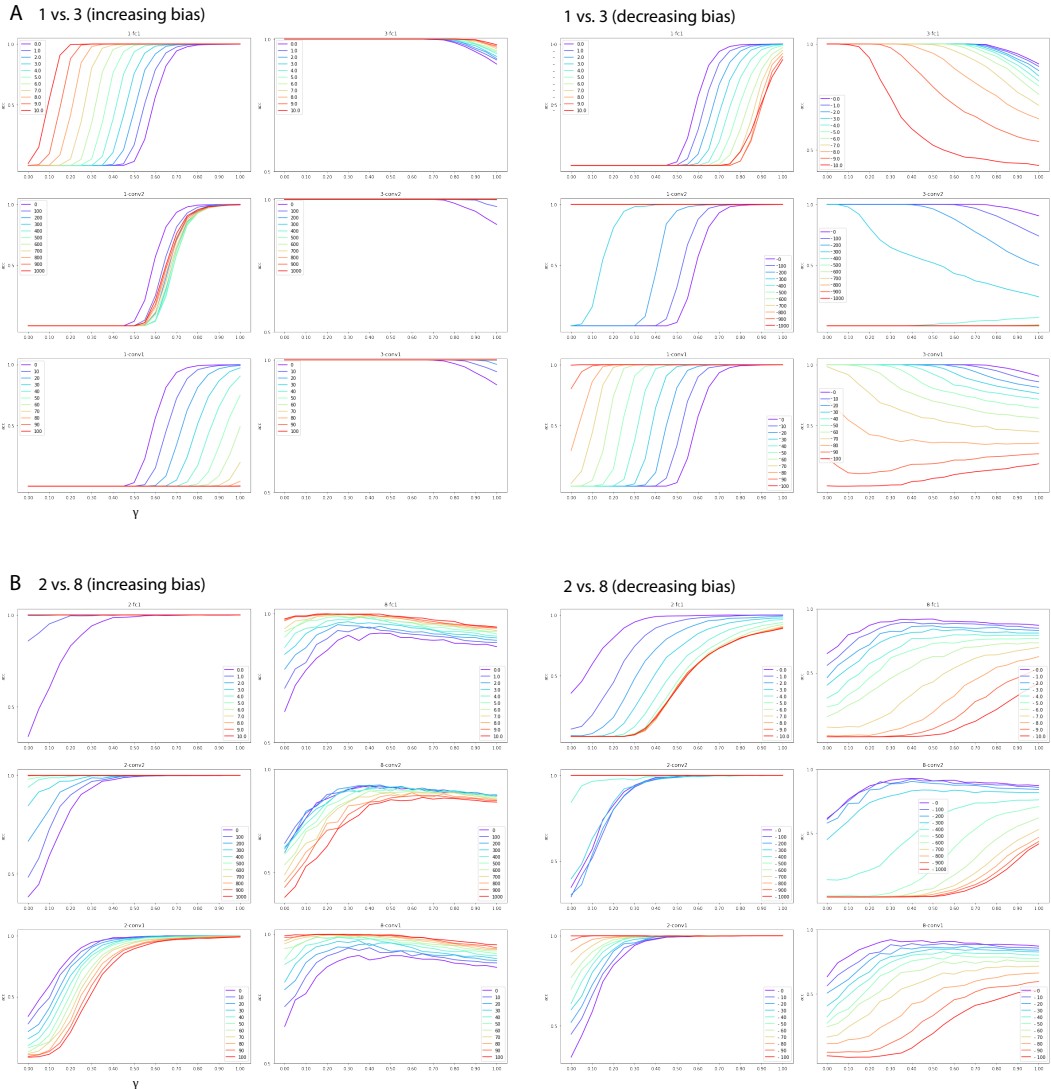

Figure 25: Results of microstimulation for a two binary decision making tasks using a CNN classifier (1 vs. 3) and (2 vs. 8). Left(right) panels show increasing (decreasing) bias for each layer. See Fig. 9 in the main text.

