# OpenReview forum: "White Noise Analysis of Neural Networks"
_ICLR.cc/2020/Conference — Accept (Spotlight)_

### Official Review · AnonReviewer2 · 2019-10-23
**Official Blind Review #2**

**Rating:** 3

**Review:**

This paper uses classification images and spike triggered averages to
reveal hidden structure in deep neural networks.  As the authors
clearly state, the techniques used are not novel (they have been used
in neuroscience) but the results are interesting.

One interesting aspect is that the classification image technique is
quite good (better than the average gradient of the classification
loss with respect to the image) at revealing trojan (adversarial
patch) networks - networks trained with trojan patches that change the
classified class of the image - (and the area of the trojan attack).
How long does it take (and with what hardware) to create the
classification images for the trojan networks?

The authors also show how the method can be used for black-box adversarial
attacks without need for gradients or logits and model microstimulation
experiments.

I don't feel there is quite enough insights here for ICLR publication.


**Experience Assessment:**

I have published in this field for several years.

**Review Assessment: Checking Correctness Of Derivations And Theory:**

I assessed the sensibility of the derivations and theory.

**Review Assessment: Checking Correctness Of Experiments:**

I assessed the sensibility of the experiments.

**Review Assessment: Thoroughness In Paper Reading:**

I read the paper at least twice and used my best judgement in assessing the paper.

---

> ### Author Response · Authors · 2019-11-10
> **Insights and computational complexity/run-time**
>
> > As the authors clearly state, the techniques used are not novel (they have been used
> in neuroscience) but the results are interesting.
>
> That is true. These are well-established sound tools in neurosciences and are quite popular. We have contributed in several ways. First, we
> conduct the first white noise analysis of deep neural networks, in fact any type of neural network, to the best of our knowledge. Second, we show that these tools are quite efficient in understanding a black box system in terms of gaining a first order approximation of its functionality, influencing its response, as well as diagnosing backdoor attacks. Third, we show, again for the first time, that deep neural networks show a similar behavior as humans during microsimulation, yet another evidence that deep networks might indeed be a decent model of the nervous system (at least in the visual domain). Our work looks at the problem from a basic science perspective and provides tangible engineering applications, perhaps not the state of the art but with high potential to improve upon. We believe these are important insights here that can drive future research in this direction and in bridging research on biological and artificial neural networks.
>
>
>
>
> >  How long does it take (and with what hardware) to create the classification images for the trojan networks?
> It takes about 9 minutes, on a single 2080Ti GPU with 11G RAM, to feed 1 million CIFAR10 RGB images (32 x 32 pixels; batch size 250) to compute the bias maps. The corresponding number for MNIST gray images (28 x 28 pixels) is about 2 minutes. Using the Gabor filter technique (section 3.1, page 5; and appendix), we were able to cut these numbers down to 25 seconds, including both PCA and bias computation steps. As is shown in Figure 14 (Supplement), 100K samples is enough to reveal the bias in a CNN. We will add and thoroughly discuss computational complexity of our approach to the paper.
>
>
> Please see also our response to the Reviewer #4 and #1.

---

### Official Review · AnonReviewer1 · 2019-10-23
**Official Blind Review #1**

**Rating:** 8

**Review:**

This is an interesting paper that uses two techniques from neuroscience (eletrophysiology) to interpret CNNs. The "classification images" technique allows the authors to build an input specific bias map for CNNs, from different images that are randomly generated using different methods. STA is also used to visualize filters. The authors find interesting similarities between CNNs and animal/human brain such as the shift of the psychometric curve. The paper is well written and the experiments are thorough.

This paper has comprehensive use of neuroscience method, nicely bringing the two fields together. Even though the authors mention other approaches, perhaps one weakness is the limited discussion of other visualization tools: what is the difference between other more frequent approaches such as activation maximization and the approach the authors present. What could we understand with these tools that cannot be understood through activation maximization (i.e. using backprop to find optimal stimuli)? One problem with classification images is that the pattern is limited to low level patterns unless the random images have structure in them. The authors solve this problem using more complex random sample (e.g. with Gabor features). But a more in depth discussion is necessarily about the additional understanding that is procured from such an approach in comparison to activation maximization. It's an important discussion because even neuroscientists are now interested (and finally able) to properly perform activation maximization to test their hypotheses more strongly (see Bashivan et al 2019 science for a great example).

Even though I think the lack of this discussion is currently a limitation, I think this paper is a good opportunity to have this discussion and bring together the specifics of these two different approaches.


**Experience Assessment:**

I have read many papers in this area.

**Review Assessment: Checking Correctness Of Derivations And Theory:**

N/A

**Review Assessment: Checking Correctness Of Experiments:**

I assessed the sensibility of the experiments.

**Review Assessment: Thoroughness In Paper Reading:**

I read the paper at least twice and used my best judgement in assessing the paper.

---

> ### Author Response · Authors · 2019-11-10
> **Further discussion on confluence of biological and artificial neural networks and future potentials**
>
> Thank you for constructive feedback.
>
> > what is the difference between other more frequent approaches such as activation maximization and the approach the authors present.
>  What could we understand with these tools that cannot be understood through activation maximization (i.e. using backprop to find optimal stimuli)?
>
> The main difference lies in the types of assumptions and available information about a system. Almost all activation maximization models assume access to the gradients as well as network response whereas here we only need the latter.
>
>
>
>  > One problem with classification images is that the pattern is limited to low level patterns unless the random images have structure in them. The authors solve this problem using more complex random sample (e.g. with Gabor features). But a more in depth discussion is necessarily about the additional understanding that is procured from such an approach in comparison to activation maximization. It's an important discussion because even neuroscientists are now interested (and finally able) to properly perform activation maximization to test their hypotheses more strongly (see Bashivan et al 2019 science for a great example).
>
>
> Bashivan et al 2019 is indeed a very interesting work.
> The technique they used is activation maximization from the original 2009 paper (Erhan et al.). They iteratively changed the pixel values in the direction of the gradient to maximize the firing rate. This requires gradient obviously. The interesting point of the paper is building the mapping between ANN and V4 neurons, so as to achieve better understanding of brain network functions using ANNs. Our goal is the opposite, namely using human visual methods to understand a blackbox ANN. We believe our approach can be useful to improve such activation maximization approaches. For example, instead of working in pixel domain, perhaps activation maximization can be applied on subtle image structures to Lower sample complexity. This is very important when dealing with human subjects in behavioral experiments or animals in electrophysiology experiments. Other than that, we believe our work carries immense potential for psychiatric applications to study the behavior of patients and diagnosis.
>
>
>
>
> > Even though I think the lack of this discussion is currently a limitation, I think this paper is a good opportunity to have this discussion and bring together the specifics of these two different approaches.
>  Thanks for the suggestion. We will indeed discuss this more thoroughly in section 4 and invite other researchers to comment on this.
>
>
>
> Reference:
> Erhan, Dumitru, et al. "Visualizing higher-layer features of a deep network." University of Montreal 1341.3 (2009): 1.

---

> > ### Author Response · Authors · 2019-11-13
> > **Further discussion on Bashivan et al., and related works**
> >
> > Bashivan et al. mention two limitations of ANNs, lack of interpretability and weak generalization.
> > They used an ANN (matched to data) to synthesize novel “controller” images based on the model’s implicit knowledge of the ventral visual stream. The ANN was used it to construct images predicted to either broadly activate large populations of neurons or selectively activate one population while keeping the others unchanged.
> >
> > In addition to activation maximization (called stretch method in their paper), Bashivan et al use another method they call "one-hot population” that tries to maximize the activity of a v4 neuron while maximally suppressing
> > activity of other neurons. The latter is reminiscent of some theories of neural coding in visual search (e.g., optimal gain theory; Navalpakkam and Itti, Neuron 2007). While both methods were effective, the synthesized images by the latter approach seem to have more structures in them.
> >
> > Some of our analysis here suggest complementary experiments to works in this real.
> > For example, we were able to find the biases in each layer of a CNN by sending noise throughout the network. We could then add these biases to a particular layer of neurons when an image was presented to the network which led to biasing the final decision of the network in some cases (See Figs 20 and 21 in the supplement). Perhaps approaches like this (i/e., adding bias to all neurons in a layer or increasing gains of individual neurons based on their activations; multiplicative gain) can be also tested in electrophysiology by using neural-data matched CNNs, akin to Bashivan et al.’ work. Further, our computational analysis on microstimulating neurons in different layers of CNNs provides an opportunity to assess how well CNNs can explain neural data, for example by using biases computed from models to drive a neuron when stimulus is augmented with noise (See Fig. 9 main text and Figs 23-25 supplement).
> >
> > We believe this path in using models and neurophysiology in a closed-loop manner is very promising in improving our understanding of deep neural networks, in particular CNNs, and holds a great promise for brain-computer interfacing. Further, our work opens an opportunity to relate several areas in computational neuroscience and deep learning (adversarial ML, visual search, neural coding, microsimulation, neural models of ventral stream, and BCI).

---

### Official Review · AnonReviewer3 · 2019-11-03
**Official Blind Review #3**

**Rating:** 6

**Review:**

Summary:
This paper introduces two tools from spike analysis to understand the bias neural networks have. The first tool is classification images and the second, spike-triggered analysis.

The broad goal of the paper is to add more tooling to add interpretability and robustness to a neural network.

Classification images can be summarized as: Produce a stimulus from the sum of a signal image and a noise image. Then average the response over many trials to determine what template the network used.

Spike-triggered analysis can be understood as measuring a neuron's response to time-varying stimuli (ie: neuron responding to a line) which scopes out the receptive field.

The paper is well written. Experimentation section is thorough. The related work is well discussed.
The overall techniques are interesting and can help the community think about interpretability by using tools from related disciplines.

Decision:
Accept

Reasoning:
Interpretability is a very important problem and the authors present ways of thinking about it from the lens of computational neuroscience. This paper has the potential to inspire future research in this direction.





**Experience Assessment:**

I have read many papers in this area.

**Review Assessment: Checking Correctness Of Derivations And Theory:**

I assessed the sensibility of the derivations and theory.

**Review Assessment: Checking Correctness Of Experiments:**

I carefully checked the experiments.

**Review Assessment: Thoroughness In Paper Reading:**

I read the paper at least twice and used my best judgement in assessing the paper.

---

> ### Author Response · Authors · 2019-11-10
> **Impact and potential for future**
>
> Thank you very much for the encouragement.
>
> > Interpretability is a very important problem and the authors present ways of thinking about it from the lens of computational neuroscience. This paper has the potential to inspire future research in this direction.
>
> We have indeed introduced new tools, borrowed from computational neuroscience, to the artificial neural networks community. These are quite interesting, powerful, and general tools with high potential for defending against adversarial attacks as well as interpreting neural networks, not only in the visual domain but also other areas.

---

### Official Review · AnonReviewer4 · 2019-11-03
**Official Blind Review #4**

**Rating:** 6

**Review:**

This work inspects the bias existing in the neural network classifiers through two techniques from computational neuroscience, the classification image techniques and spike-triggered covariance (STA) analysis. The classification image is generated as the difference map between all the averaged images with predicted classes under noise perturbation. STA is used to generate the eigenvector as the input and response, as a visualization. Both of the tools are the standard psychophysics tools.

The authors use the two tools to visualize the bias learned by various classifiers, CNN, MLP, and logistic regression, trained on three datasets MNIST, Fashion-MNIST and CIFAR 10. There are interesting patterns that emerged from the classifiers trained on MNIST and Fashion-MNIST by the classification image technique, but the result from the CIFAR 10 is not explainable. The classification image is further used to analyze the adversarial attack and defense while STA is used to visualize the internal units.

Strengths:
The paper is well-written and easy to follow. With detailed and complete psychophysics experiments, several interesting properties of the filters and the biases of the neural network classifiers are revealed. It is great to see that more tools from psychophysics can be applied to understand neural network classifiers.

Weakness:
1. One concern I have is that the result on CIFAR-10 is not interpretable. See figure 3. On the other hand, the visualization of filters in higher layers (conv4 and conv6) from the net trained on CIFAR-10 in Figure 8 still looks like edge detectors. There are not so many discussions and explanations on what possibly happened here.

2. The inferior result on CIFAR-10 might indicate that the techniques cannot be generalized to interpret large-scale networks such as AlexNet or resnet, trained on ImageNet or Places. It is 2019 now, I expect to see more experiments on at least AlexNet or VGG, rather than tiny sets MNIST or CIFAR. I will suggest the authors run experiments on AlexNet trained on ImageNet/Places, then we can compare the visualization with the other filter visualization to see the difference.

3. There is no comparison for other bias visualization methods. Following the idea of mean images, I think you can also show the mean images conditional on the predicted labels (average all the testing images with certain predicted labels). How the result from the psychophysics is different from that? How well the proposed methods are able to identify the bias? What other bias cannot be identified? Some discussions on the failure cases would be appreciated


**Experience Assessment:**

I have published in this field for several years.

**Review Assessment: Checking Correctness Of Derivations And Theory:**

I assessed the sensibility of the derivations and theory.

**Review Assessment: Checking Correctness Of Experiments:**

I assessed the sensibility of the experiments.

**Review Assessment: Thoroughness In Paper Reading:**

I read the paper at least twice and used my best judgement in assessing the paper.

---

> ### Author Response · Authors · 2019-11-10
> **Limitations of the approach**
>
> We thank the reviewer for the constructive feedback.
>
> We have scratched the surface of possibilities for employing tools from computational neuroscience and psychology for machine learning and computer vision, especially for understanding and improving ANNs (it is actually a two-way street!). We will further discuss other tools, techniques, and ideas that researchers can borrow from neuroscience to improve deep learning models (e.g., normalization techniques, short and long range dependencies; Linsley et al).
>
> > 1. One concern I have is that the result on CIFAR-10 is not interpretable ...
>
> Notice that since we generate the white noise input completely random for all three color channels, the resulting average is inevitably jittered. Averaging images also does not represent objects’ higher level structures, especially when the data is not calibrated. Although the mean noise maps are not visually interpretable, they do contain useful information that can used for classification and attack diagnosis.
> The higher layers are visualized through averaging the input white noise that triggered that particular filter, instead of giving real images. That is why visualizations in Figure 8 resemble lower level features. STA can not fully characterize the type of stimuli a neuron in mid and higher layers responds to. It is average response. Other techniques such as spike triggered covariance aim to find dimensions to which a neuron responds to, akin to visualization methods in ANNs (e.g., Zeiler et al., 2014).
>
> This is the major limitation of these techniques and is well-known in the literature. It is in fact the main reason why they have not been widely adopted to natural scenes. Here, we tried to mitigate this limitation by generating noise patterns with faint strutters. We were able to gain significantly better results using much less number of samples. See Figure 4.
>
> Nonetheless, despite these limitations, these tools are very effective, in particular when we other options are off the table (i.e., no assumptions can be made). This makes them quite appealing to inspect black-box adversarial attacks. We will add these comments and more discussions to the main text.
>
> > 2. The inferior result on CIFAR-10 might indicate  ...
>
> Objects in datasets like ImageNet are uncentered and uncalibrated, which will result in losing too much information through averaging, making them unsuitable to apply this particular technique.
>
> As we discussed above, we do not expect to obtain bias maps that look quite like target classes (using STA). Why not? Because even averaging the real samples does not lead to anything visually meaningful! Nonetheless, the computed bias maps contain rich information that can be used for classification and also to inspect models. For example, a model that performs close to random will not be able to generate bias maps that are informative. On the other hand, a perfect classifier will generate bias maps equal to mean images (i.e.,upper-bound is the mean images). As a result, we do not expect the bias maps to improve much with AlexNet or even ResNet beyond what mean images already provide. What we aimed to convey was mainly the point that bias maps indeed contain significant information and are useful for a number of purposes.
>
> > 3. There is no comparison for other bias visualization ...
>
> This is what we have actually done. There is a subtlety here, though. In psychophysics experiments, usually signal is embedded in white noise and tasks are usually binary. The main reason is to lower the number of trials. For artificial classifiers, however, we do not have this limitation. Thus, we can bombard a classifier with tons of just noise patterns in large scale (even multi-way classifiers).
>
> The mean images we showed are conditioned on the predicted labels, namely we average noise that are predicted into a certain class to get the classification image of that class. We believe for well-calibrated datasets like MNIST and FashionMNIST, the results are very similar to psychophysics results: class difference are highlighted and common parts are disregarded (darker color). In neuroscience experiments, the designed stimuli are also often centered and calibrated. Higher-level biases (e.g., semantic level instead of shape level) might not be detected.
>
> Also, there are really not many ways to pinpoint the biases of neural networks. We would be happy to try new ones if you can suggest some.
>
> Regarding the limitations and failures of the approach, we have clearly stated them in the paper in several occasions. For example, the method has a high sample complexity to work well but can be mitigated to some extent. At the end of the day these tools have some pros and cons, but are certainly a nice addition to the deep learning toolbox.
>
> Reference:
> -Linsley,et al., Learning long-range spatial dependencies with horizontal gated recurrent units, NIPS 2018
> -Zeiler, et al. Visualizing and understanding convolutional networks. ECCV.  2014.

---

> > ### Author Response · Authors · 2019-11-12
> > **Results on ImageNet!**
> >
> > We conducted new experiments to test the proposed methods on large scale recognition datasets. ImageNet validation set including 50K images covering 1000 categories and 1M samples using Gabor PCA sampling and deep CNNs pretraiend on ImageNet train set, were utilized.
> >
> >
> > As results in the table below show, even with 1M samples, and without parameter tuning, we can get an improvement over the chance level  which is 0.0010 or 0.1%. We obtain 2x better accuracy than the chance using ResNet152. It seems that 1M samples is not enough to cover all classes. For example, no noise pattern gets classified under almost half of the classes using ResNet152. For some other models, even a larger number of classes remain empty (yet another evidence that white noise can reveal biases in models). We believe it is possible to improve these results with more samples. As you can see with more number of classes filled, a better accuracy can be achieved. It takes only few minutes (about 2) to process all 1M images at resolution 32 x 32. Notice that ImageNet models have been trained on 224 x 224 images, while here we test them on 32 x 32 images for the sake of computational complexity. A better way would be to train the models on 32 x 32 images or feed the noise at 224 x 224 resolution. This may takes more time but may result in better performance.
> >
> > Regarding the generated filters, we found that the same types of edge detectors emerge in the early layer of CNNs just like the ones we observed over CIFAR10 dataset.
> >
> >
> > Backbone		Accuracy	       Run Time	    Number of classes with 0 samples
> > ----------------------------------------------------------------------------------------------------------------------------
> > ResNet152 		0.00180	    		2:15 mins           564
> > ResNet101 		0.00152	    		1:36 mins           539
> > densenet201		0.00118	    		2:12 mins           998
> > squeezenet1_1	0.00104	    		0:13 mins           999
> > googlenet	        0.00102	    		0:26 mins           999
> > mnasnet1_3          0.00082			0:32 mins           922
> > vgg_19_bn		0.00074			1:51 mins           994
> >
> >
> >
> >
> > Overall, our pilot investigation on large scale datasets are promising. We believe better results than the ones reported above are possible with further modifications (e.g., using better distance measures between an image and the average noise map for each class). Also, it is highly likely that increasing the number of samples (white noise inputs) will lead to better performance.
> >
> >
> > We will add these results in the paper and will discuss them.

---

> > > ### Comment · AnonReviewer4 · 2019-11-13
> > > **Thanks**
> > >
> > > Thanks for the response and for adding the result on ImageNet. I will keep my score as weak accept and support the acceptance of this work.
> > >
> > > I think this paper has an interesting perspective and many solid experiments to justify the existence of bias. This paper could be a good contribution to ICLR'20.

---

### Decision · Program_Chairs · 2019-12-19

**Decision:**

Accept (Spotlight)

**Comment:**

All the reviewers found the paper to contain an interesting idea with insightful experiments. The rebuttal further improved confidence of the reviewers. The paper is accepted.